# High-parametric protein maps reveal the spatial organization in early-developing human lung

Sanem Sariyar[1,2], Alexandros Sountoulidis[1,3], Jan Niklas Hansen [1,2,4], Sergio Marco Salas [1,5], Mariya Mardamshina[1,2], Anna Martinez Casals[1,2,4], Frederic Ballllosera Navarro[1,2,4], Zaneta Andrusivova[1,6], Xiaofei Li [7], Paulo Czarnewski[1,6], Joakim Lundeberg [1,6], Sten Linnarsson [8], Mats Nilsson [1,5], Erik Sundström [7], Christos Samakovlis[1,3,9], Emma Lundberg [1,2,4,10,11] ✉ & Burcu Ayoglu [1,2,11] ✉

The respiratory system, including the lungs, is essential for terrestrial life. While recent research has advanced our understanding of lung development, much still relies on animal models and transcriptome analyses. In this study conducted within the Human Developmental Cell Atlas (HDCA) initiative, we describe the protein-level spatiotemporal organization of the lung during the first trimester of human gestation. Using high-parametric tissue imaging with a 30-plex antibody panel, we analyzed human lung samples from 6 to 13 post-conception weeks, generating data from over 2 million cells across five developmental timepoints. We present a resource detailing spatially resolved cell type composition of the developing human lung, including proliferative states, immune cell patterns, spatial arrangement traits, and their temporal evolution. This represents an extensive single-cell resolved protein-level examination of the developing human lung and provides a valuable resource for further research into the developmental roots of human respiratory health and disease.

The vital gas exchange between our body and inhaled air primarily occurs across the lung's extensive surface area, which also serves as a physical barrier against inhaled pathogens, allergens and toxins[1]. Physiologic respiration relies on the structural and functional integrity of the two highly branched networks of epithelial and endothelial cells, which are tightly juxtaposed in the distal lung compartment to create the gas exchange surface of the alveolar respiratory units. These networks are surrounded and supported by various stromal (mesenchymal) cell types and complemented by resident and circulating immune cells playing a critical role in recognition of pathogens and activation of host defense mechanisms[2,3]. Various types of neurons also innervate the lung, sensing environmental changes and regulating essential functions such as breathing rhythm[4] and bronchoconstriction[5].

Recent single-cell transcriptomics studies have identified approximately 60 distinct cell types and states in the non-diseased adult human lung[6–8], with additional ones associated with disease[9,10].

[1]Science for Life Laboratory, Solna, Sweden. [2]Department of Protein Science, KTH—Royal Institute of Technology, Stockholm, Sweden. [3]Department of Molecular Biosciences, Wenner-Gren Institute, Stockholm University, Stockholm, Sweden. [4]Department of Bioengineering, Stanford University, Stanford, CA, USA. [5]Department of Biochemistry and Biophysics, Stockholm University, Stockholm, Sweden. [6]Department of Gene Technology, KTH—Royal Institute of Technology, Stockholm, Sweden. [7]Department of Neurobiology, Care Sciences and Society, Karolinska Institutet, Stockholm, Sweden. [8]Department of Medical Biochemistry and Biophysics, Karolinska Institute, Stockholm, Sweden. [9]Molecular Pneumology, Cardiopulmonary Institute, Justus Liebig University, Giessen, Germany. [10]Department of Pathology, Stanford University, Stanford, CA, USA. [11]These authors contributed equally: Emma Lundberg, Burcu Ayoglu. ✉e-mail: emma.lundberg@scilifelab.se; burcu.ayoglu@scilifelab.se

Intriguingly, this cellular diversity arises from the endoderm-derived foregut, which forms the epithelial network[11]; the surrounding meso-derm, which produces all stromal cell types including pulmonary vasculature[12–15]; and ectoderm-derived neural crest progenitors, which generate parasympathetic postganglionic neurons that innervate the airways[16,17]. Immune cell progenitors also colonize the lung early in development, establishing lung resident immune cells[18,19].

Our understanding of lung cell differentiation and maturation derives mainly from studies using animal models and in vitro systems[20,21], which have been instrumental in characterizing evolutio-narily conserved developmental mechanisms and identifying numer-ous key signaling pathways and transcription factors in lung development[11]. However, biologically significant differences exist between animal and human lung physiology[22], and in vitro induced pluripotent cell models show limited ability to recapitulate the com-plex cellular interactions involving cells across different germ layers[23]. Therefore, studying intact human embryonic lung tissues is essential for improving our understanding of lung formation, and in turn, to advance strategies for preventing and treating respiratory diseases[24,25].

Despite the limited availability of human embryonic tissues, recent initiatives such as the Human Developmental Cell Atlas (HDCA)[26] have enabled us[27] and others[28–30] to perform detailed single-cell tran-scriptomic and spatial transcriptomic characterization of over 80 dif-ferent cell types and states during early human lung development. These studies illuminated the dynamic processes involved in human lung development and provided important insights into the spatial organization of cells in the developing human lung at transcriptome level. However, to our knowledge, no comprehensive studies have yet mapped the single-cell resolved spatial organization of key cell types in the developing human lung at the protein level. As proteins are the effectors of biological processes, knowledge of their distribution and relative abundance within different cell types and tissue regions can offer functional insights[31] that may not be evident from spatial tran-scriptomics data alone, or from single-cell transcriptomics data of dis-sociated cells, where valuable spatial information is per se lost.

To address this gap, we conducted a spatially resolved protein analysis of the developing human lung during the first trimester of gestation. We developed a panel of oligo-barcoded antibodies speci-fically validated to visualize the key cell types in the developing human lung and used high-parametric microscopy for simultaneous analysis of 30 proteins across sections of human prenatal lung tissue. The 30-plex antibody panel enabled the generation of protein data from over 2 million single cells across five distinct time points ranging between 6 and 13 post-conception weeks (pcw). Our high-parametric protein expression analysis revealed the spatial organization of the main cell types of the developing human lung, tracked their relative abundances and proliferation dynamics over time, and identified consistent adja-cency patterns between different cell types, including a previously unrecognized enrichment of immune cells near the developing arterial endothelium. This single-cell resolved protein-level imaging resource is openly accessible through an online platform (https://hdca-sweden. scilifelab.se/tissues-overview/lung/), enabling further interactive exploration of spatial data from human whole lung tissue sections during first trimester of gestation.

## Results
### Establishment of a multiplexed antibody panel to characterize the developing human lung microenvironment
To characterize the cellular heterogeneity and spatial organization of developing human lung tissue, we evaluated 72 antibodies that could potentially be used in a multiplexed antibody panel (Supplementary Table 1, Supplementary Fig. 1). We tested each antibody in three con-secutive validation steps: (1) indirect immunofluorescence staining (IF); (2) validation of the oligo barcode-conjugated antibody; (3) high-parametric imaging (Fig. 1A). We excluded antibodies that did not yield

a sufficient signal to noise ratio (SNR) and/or showed unspecific or no staining patterns. After these stringent validation rounds, 30 of the 72 antibodies were selected to be included in the final antibody panel and their SNRs were optimized further by titrating the antibody dilution factors (Fig. 1B, Supplementary Table 1). Among the antibodies that were excluded, 43% were excluded after initial IF screening due to non-specific staining patterns, 19% were excluded after barcode-conjuga-tion, and 38% were excluded after evaluation of staining performance in optimization runs as part of the multiplexed panel. These results highlight the increased complexity of multiplexed antibody assays and emphasize the importance of validation efforts.

The panel design was based on cell type markers proposed in recent single-cell transcriptomic studies of human adult[6,32] and developing lung[27] (Fig. 1C, Table 1). We included general epithelial markers, such as EPCAM (Epithelial cell adhesion molecule), PanCK (Pan Cytokeratin), CDH-1 (Epithelial cadherin, E-Cadherin), TTF1 (Transcription termination factor 1, also known as Nkx2.1), as well as SOX2 and SOX9 (SRY-box transcription factor 2 and 9) in the panel to map the proximal and distal epithelial cells[21]. For endothelial cells, we incorporated CD31, CD34, CD123, CD144 and CLDN5 (Claudin 5) as general endothelial markers, PDPN (Podoplanin) as a lymphatic endothelial and mesothelial marker and PRX (Periaxin) as a capillary marker. We added markers such as VIM (Vimentin), DCN (Decorin), and COL1A1 (Collagen Type I) to map the mesenchymal cells, and ACTA2 to map smooth muscle cells. In addition to the general immune markers of CD45 and HLA-DR (Human leukocyte antigen DR isotype), we included other markers to further delineate immune cells, such as CD3 and CD4 for T cells and CD68, CD163 and MRC1 (Mannose receptor C-type 1) for macrophages, and CD19 for B cells. We also included Ki67 to identify proliferative cell states (Fig. 1C).

### Spatially resolved protein-level cell type composition of the developing human lung
Using the validated multiplexed antibody panel, we performed tissue imaging on 6-, 8.5-, 11-, 12-, and 13-pcw-old human lung tissue sections (Fig. 1D–F, Supplementary Fig. 2). For each week, we segmented the images using DAPI as a nuclear marker and EPCAM as a membrane marker (Supplementary Fig. 3), and selected artifact-free regions for downstream analysis (Supplementary Fig. 4A, B). Cellular density was generally consistent across the developmental weeks, with a slight increase observed in the later weeks (Supplementary Fig. 4C).

As summarized in the flowchart in Supplementary Fig. 5, the resulting single-cell resolved protein expression data from each developmental week was first clustered to identify and remove remaining artifact regions (Supplementary Fig. 6). To gain an initial insight into the main cell types identifiable in human lung tissue from this developmental timeframe, data from all developmental weeks was then merged (Supplementary Fig. 7A), the identified cell type clusters were annotated (Supplementary Fig. 7B), and clusters belonging to the same major cell type were combined for a broader annotation (Sup-plementary Fig. 8A). As shown in Supplementary Fig. 8B, the expres-sion of markers such as SOX2, SOX9, TTF1, EPCAM, and E-cadherin was high in epithelial cell clusters. Also, CD44 was expressed in both SOX2$^{high}$ (proximal) and SOX9$^{high}$ (distal) epithelial cells, with relatively higher expression in the latter cell cluster. In endothelial and lymphatic endothelial cell clusters, relatively higher levels of CD31, CD144, and CD90 were observed. Lymphatic endothelial cells exhibited lower levels of CD123 than endothelial cells, but higher levels of PDPN. As summarized in Supplementary Fig. 8C, the most abundant cell type in the developing human lung was mesenchymal, accounting for 33.9% of the cell population, followed by endothelial cells (24.6%), SOX2$^{high}$ epithelial cells (13.1%), SOX9$^{high}$ epithelial cells (9.7%), airway smooth muscle cells (9.2%) immune cells (4.8%), lymphatic endothelial cells (2.1%), neuronal cells (1.1%), vascular smooth muscle cells (0.8%) and chondroblasts (0.7%).

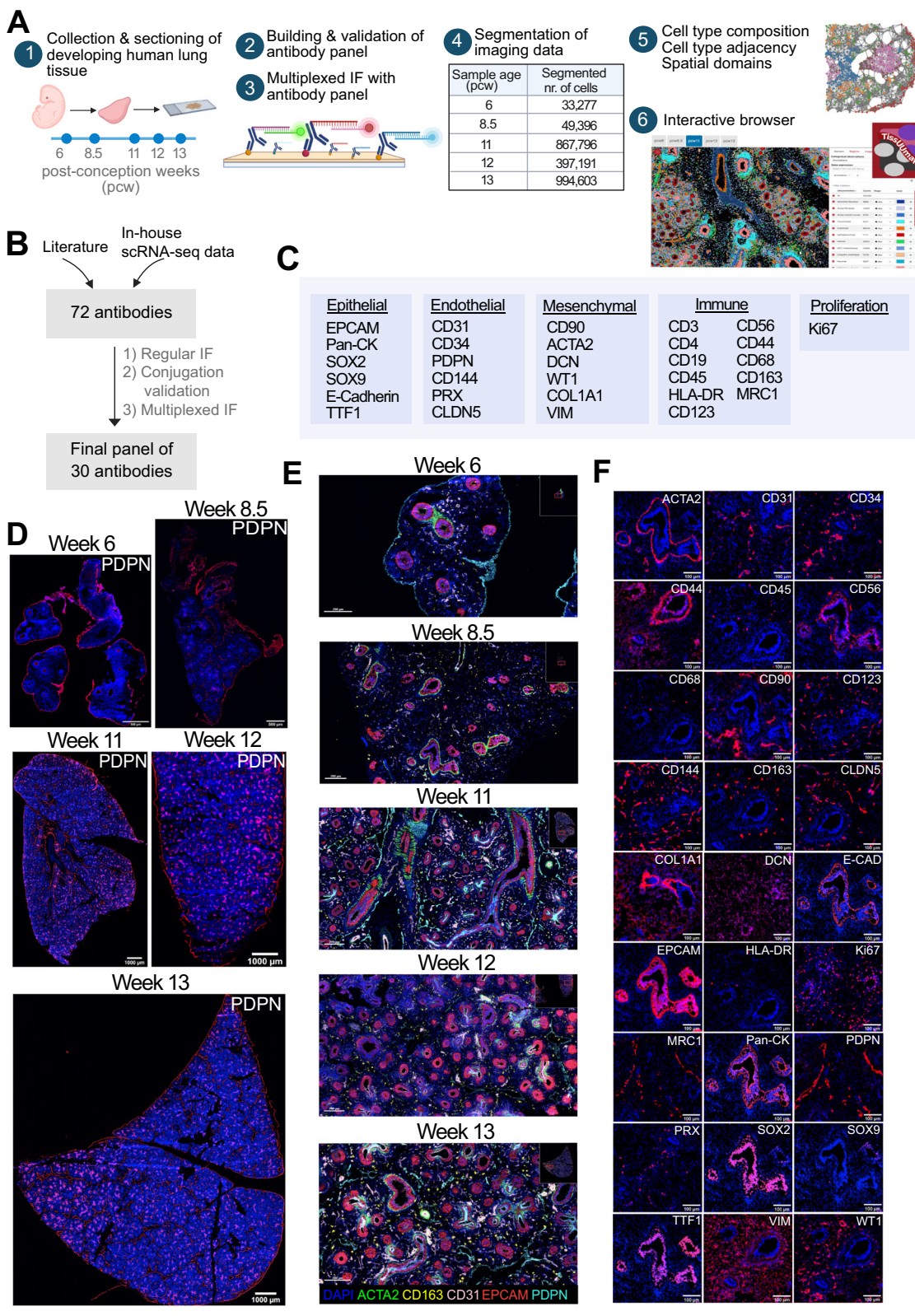

Next, the data clustered individually for each week was re-annotated for a more detailed analysis of the spatial composition and neighboring patterns of the major cell types at each developmental week and how these change over the course of early lung development (Supplementary Figs. 9–11). The cell type annotation for each cell was visualized in scatterplots to assess and confirm microanatomic consistency (Fig. 2A). Endothelial structures had already formed at week 6, while continuous endothelial structures appeared from week 8.5 onwards (Fig. 2D). Lymphatic endothelial structures were not observed at week 6 but were present in tissue sections from all later stages (Fig. 2E, Supplementary Fig. 12). Immune cells were observed starting from week 6 (Fig. 2A). The expression patterns of each marker across the annotated cell types (Fig. 2B) as well as correlations between the markers (Supplementary Fig. 13) agreed with previous single-cell transcriptomics reports[27,28].

**Fig. 1 | Multiplexed imaging and spatial characterization of developing human lung tissue. A** Tissue samples from 6- to 13- post conception week (pcw)-old human lungs were sectioned. Each selected antibody was tested for immunofluorescence (IF) staining, and then conjugated with DNA barcodes and validated. High-parametric imaging was performed, and imaging data was segmented for downstream analysis to characterize the cell type composition, cell type adjacency patterns, spatial domains and their temporal changes in the developing human lung. Annotated cell types and expression patterns for each of the used 30 antibodies overlaid on DAPI channel images were presented in an openly accessible, interactive portal at https://hdca-sweden.scilifelab.se/tissues-overview/lung. (Created in BioRender. Ayoglu, B. (2024) BioRender.com/t97n413). **B** A total of 72 antibodies, selected based on literature and in-house single-cell transcriptomics data[27], underwent comprehensive validation steps. Ultimately, 30 of these were selected to be included in the final panel used for multiplexed imaging of developing human lung tissue. **C** The content of the 30-plex antibody panel used for multiplexed imaging. **D** Whole tissue images of lung samples at week 6–13 showing PDPN in red, DAPI in blue. In week 6, PDPN expression is observed only in mesothelium and from week 8.5 onwards, it is observed also in lymphatic endothelium, and epithelium and mesothelium. Images represent a single donor for each developmental week. Scale bars for weeks 6 and 8.5 correspond to 500 μm, and for weeks 11 to 13 to 1000 μm. **E** Representative images for five antibodies overlaid to show the main structures and cell types of the developing human lung: ACTA (smooth muscle marker) in green; CD163 (macrophage marker) in yellow; CD31 (endothelial marker) in pink; EPCAM (epithelial marker) in red; PDPN (lymphatic endothelial marker) in cyan, and DAPI (blue). Images represent a single donor for each developmental week. Scale bars correspond to 200 μm. **F** Representative staining patterns of all panel markers in selected regions of the 8.5-week-old lung (except CD3, CD4 and CD19). DAPI in blue. Scale bars correspond to 100 μm.

## Table 1 | Content of the 30-plex antibody panel built for protein profiling in developing human lung tissue

|  | Antibody-barcode | Targeted protein | In-house conjugation | Targeted main cell type/state |
|---|---|---|---|---|
| 1 | CD3-BX015 | CD3D, CD3E, CD3G | No | Immune (T-cell, Innate lymphoid cell) |
| 2 | CD4-BX021 | CD4 | No | Immune (T-cell) |
| 3 | CD19-BX003 | CD19 | No | Immune (B-cell) |
| 4 | CD31-BX032 | CD31/PECAM1 | No | Endothelial |
| 5 | CD34-BX035 | CD34 | No | Endothelial |
| 6 | CD45-BX001 | CD45/PTPRC | No | Immune |
| 7 | CD90-BX022 | CD90/THY1 | No | Mesenchymal (Fibroblast) |
| 8 | Podoplanin-BX023 | PDPN | No | Lymphatic endothelial, mesothelial |
| 9 | HLA-DR-BX026 | HLA-DRA, HLA-DRB1,... | No | Immune (Antigen presenting cell) |
| 10 | Ki67-BX047 | MKI67 | No | Proliferation |
| 11 | Pan-Cytokeratin-BX019 | KRT8, KRT18,... | No | Epithelial |
| 12 | E-Cadherin-BX014 | CDH1 | No | Epithelial |
| 13 | EPCAM-BX042 | EPCAM | Yes | Epithelial |
| 14 | CD123-BX054 | CD123/IL3RA | Yes | Immune & Endothelial |
| 15 | SOX2-BX024 | SOX2 | Yes | Epithelial |
| 16 | CD144-BX016 | CD144/CDH5 | Yes | Endothelial |
| 17 | ACTA2-BX028 | ACTA2 | Yes | Mesenchymal (Smooth muscle) |
| 18 | CD68-BX010 | CD68 | Yes | Immune (Macrophage) |
| 19 | CD44-BX020 | CD44 | Yes | Immune |
| 20 | DCN-BX013 | DCN | Yes | Mesenchymal |
| 21 | WT1-BX006 | WT1 | Yes | Mesenchymal |
| 22 | CD163- BX005 | CD163 | Yes | Immune (Macrophage) |
| 23 | COL1A1-BX002 | COL1A1 | Yes | Mesenchymal |
| 24 | CD56-RX029 | CD56/NCAM1 | Yes | Immune (NK cell) |
| 25 | MRC1-BX030 | MRC1 | Yes | Immune (Macrophage) |
| 26 | Vimentin-BX045 | VIM | Yes | Mesenchymal |
| 27 | PRX-BX052 | PRX | Yes | Endothelial |
| 28 | CLDN5-BX041 | CLDN5 | Yes | Endothelial |
| 29 | SOX9-BX033 | SOX9 | Yes | Epithelial |
| 30 | TTF1-BX007 | TTF1/NKX2-1 | Yes | Epithelial |

Information sources for targeted cell types: Nikolić et al. [21], for PDPN; Jambusaria et al. [60], for CD123; Ma et al. [61], for SOX9, Yatabe et al. [62], for TTF1. The source for other targets is Sountoulidis et al. [27] and Schupp et al., 2020[32]. Supplementary Table 1 provides information for additional antibodies evaluated to build this antibody panel.

Mesenchymal cells were the most abundant cell type in each week (Fig. 2C) and the highest fraction of them was observed at week 6 (57.7% of the total cell population). Endothelial cells were the second most abundant cell type, with an increasing trend during development (8.5% at week 6 and on average 20.8% between weeks 8.5 and 13). Endothelial cells were followed by SOX9[high] epithelial cells, SOX2[high] epithelial cells, airway smooth muscle cells and immune cells. Immune cells reached their highest abundance in week 11 and 12, constituting 7.2% and 7.3% of the total cell population, respectively. SOX2[high]

epithelial cells reached their highest abundance in week 12 and 13 (9.4% and 9.6%, respectively). Similarly, SOX9[high] epithelial cells reached their highest abundance in week 12 (15.9%). Lymphatic endothelial cells (Fig. 2E, Supplementary Fig. 12), which were not identified in week 6, were present in week 8.5, peaked in week 11 and their abundance decreased to 1.4% and 1% in weeks 12 and 13, respectively (Fig. 2E, Supplementary Fig. 12).

As discussed in the Supplementary Information, a few cell type clusters depicted in Fig. 2A, B were not consistently represented across

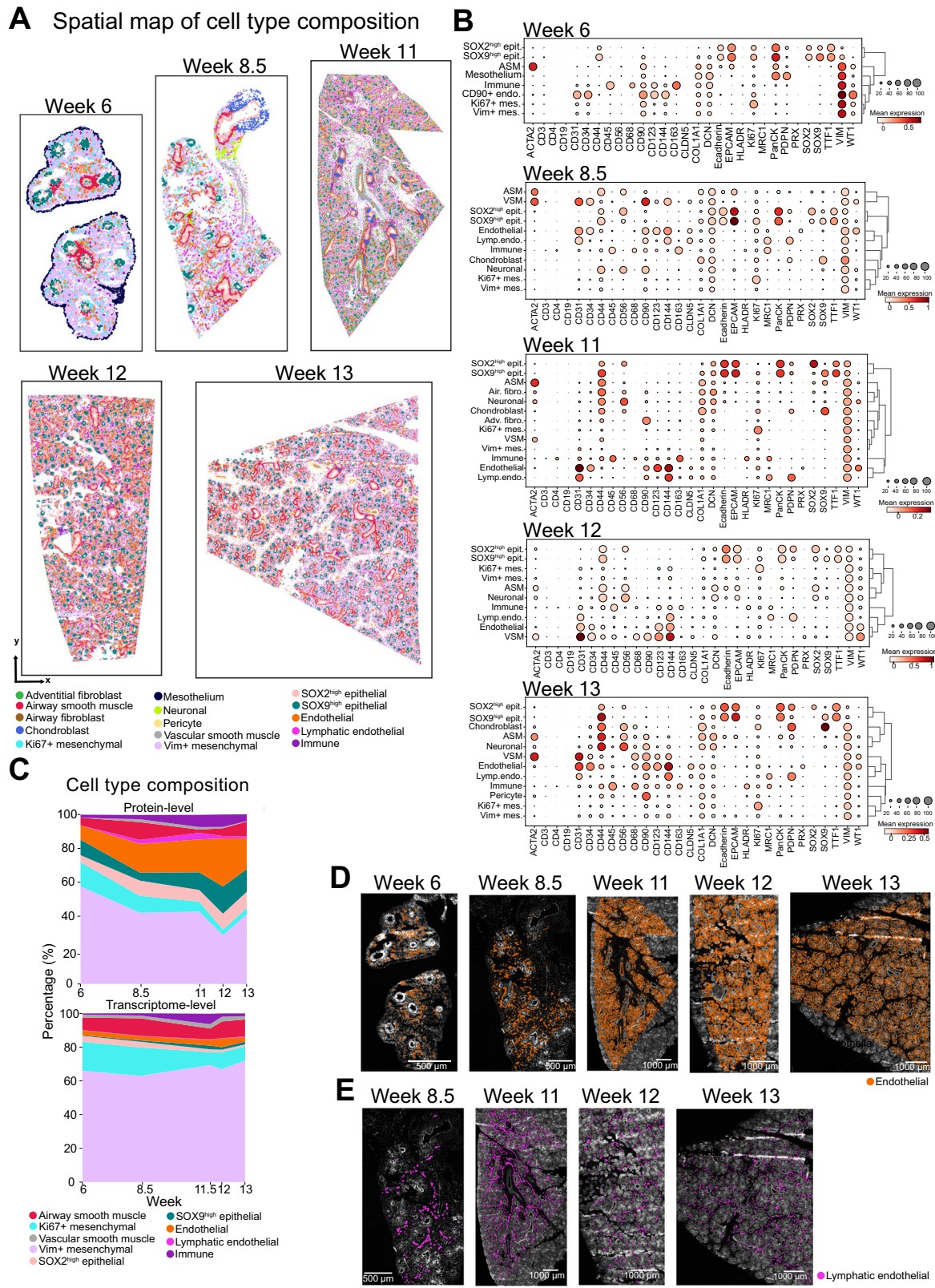

**A** Spatial map of cell type composition

Week 6, Week 8.5, Week 11, Week 12, Week 13

Adventitial fibroblast, Airway smooth muscle, Airway fibroblast, Chondroblast, Ki67+ mesenchymal, Mesothelium, Neuronal, Pericyte, Vascular smooth muscle, Vim+ mesenchymal, SOX2^high epithelial, SOX9^high epithelial, Endothelial, Lymphatic endothelial, Immune

**B** Week 6, Week 8.5, Week 11, Week 12, Week 13

**C** Cell type composition — Protein-level, Transcriptome-level

Airway smooth muscle, Ki67+ mesenchymal, Vascular smooth muscle, Vim+ mesenchymal, SOX2^high epithelial, SOX9^high epithelial, Endothelial, Lymphatic endothelial, Immune

**D** Week 6, Week 8.5, Week 11, Week 12, Week 13 — Endothelial

**E** Week 8.5, Week 11, Week 12, Week 13 — Lymphatic endothelial

all weeks due to a number of technical reasons: Uneven signal intensities in the peripheral regions of the tissue sections necessitated the exclusion of these regions from downstream data analysis. Consequently, mesothelial cell type clusters were absent from 8.5 week onwards (Supplementary Fig. 4A, B), despite their presence in all weeks as shown in Fig. 1D. Chondroblast clusters were also absent in the 12-week-old lung sample as the section presented for the 12-week-old sample originated

from a relatively more distal region of the organ compared to other samples. Similarly, adventitial fibroblast clusters, shown to localize within the bronchovascular bundles of the lung[27], were identified only in the 11-week-old sample originating from a central section of the organ, and were absent in the 12- and 13-week-old human lung samples.

Overall, our single-cell protein-level findings presented a cell type distribution that aligned with the patterns identified in the single-cell

**Fig. 2 | Characterization of main cell types in developing human lung tissue.**
**A** Spatial scatterplots illustrating the cell type composition across each week. Each dot represents a single-cell colored according to its annotated cell type after clustering of segmented imaging data from tissue regions shown in Supplementary Fig. 4A. Cell type color assignments are displayed at the bottom. **B** Dot plots displaying the average marker expression levels within each cell type cluster depicted in the selected tissue regions in panel (**A**). Circle size indicates the percentage of cells expressing the marker in each cluster. Dark red color indicates high and white color indicates low expression. ASM airway smooth muscle, VSM vascular smooth muscle, Adv. fibro adventitial fibroblast, Air. fibro airway fibroblast, Lymp. endo lymphatic endothelial. **C** Percentage of main cell types identified in the present multiplexed protein imaging dataset and a previously reported single-cell transcriptomics dataset[27]. Cell type color assignments are displayed at the bottom. Source data for this figure is provided within the Source Data file. **D** Endothelial cluster (orange) in each developmental week overlaid on DAPI channel images. Images represent a single donor for each developmental week. Scale bars correspond to 500 µm for week 6 and 8.5, and 1000 µm for week 11, 12 and 13.
**E** Lymphatic endothelial cluster (pink) in developmental weeks 8.5, 11, 12 and 13 overlaid on DAPI channel images. Images represent a single donor for each developmental week. Scale bars correspond to 500 µm for week 6 and 8.5, and 1000 µm for week 11, 12 and 13.

transcriptomics data reported by Sountoulidis et al. [27], where mesenchymal cells emerged as the most abundant cell type, while lymphatic endothelial cells were the least abundant cell type across all the weeks (Fig. 2C). Notably, our spatial protein dataset revealed higher proportions of epithelial, endothelial and lymphatic endothelial cells when compared with the single-cell transcriptomics data. These differences can be attributed to tissue dissociation bias inherent in the single-cell transcriptomics approach, leading to an underrepresentation of these cell types. Consequently, this underscores the benefit of spatial omics methodologies in offering a less biased representation of the cell type composition within the tissues.

## Cell type adjacency patterns in early human lung development

Next, we aimed to characterize the cell type adjacency patterns in the developing human lung as the communication between neighboring cells orchestrates the development of the organ. Analysis of the enrichment of cellular adjacency patterns consistently showed cells predominantly neighbored with cells of their own type across all weeks (Fig. 3B, Supplementary Fig. 14). Notably, endothelial, immune, and mesenchymal cells exhibited more frequent neighboring among each other across all the weeks (Fig. 3B). Compared to other cell types, mesenchymal cells, including proliferating mesenchymal cells, neighbored more frequently with cells of other types, indicating their dispersed distribution throughout the tissue space during the early stages of human lung development (Supplementary Fig. 14).

Examining neighboring patterns beyond homotypic ones, we observed that airway smooth muscle cells were neighboring more frequently with SOX2$^{high}$ proximal epithelial cells than with SOX9$^{high}$ distal epithelial cells from week 8.5 onwards (Supplementary Fig. 14). This spatial arrangement reflects the microanatomy of developing airways, where smooth muscle cells surround only SOX2$^{high}$ proximal epithelium, highlighting their specialized role in guiding branching morphogenesis[33]. Additionally, neuronal cells identified in our dataset from week 8.5 onwards exhibited increased adjacency with airway smooth muscle cells compared to their adjacency patterns with other cell types (Fig. 3B). In contrast, endothelial-epithelial, and mesenchymal-epithelial cell type pairs displayed the least degree of adjacency (Fig. 3B, Supplementary Fig. 14). Overall, the analysis of adjacency patterns within and between cell types provided a detailed, protein-level depiction of the emerging complex architecture of the human lung during the first trimester of gestation.

## Evolution of spatial domains during early human lung development

Tissue function depends on multi-cellular units, termed as "domains" herein, which consist of higher-order interactions involving one or more cell types. In addition to analyzing the enrichment of adjacency patterns of the cell types we annotated, we conducted an unsupervised analysis of the tissue topography by inferring the presence of distinct spatial domains. These domains, comprising one or more of the annotated cell types, were identified solely based on the spatially aggregated expression values in the tissue, i.e. independent of cell type annotations. This analysis revealed an average of six distinct spatial domains in developing human lungs, ranging between five to eight across different developmental weeks, which we annotated based on the cell type composition in each domain (Fig. 4).

Mesenchymal cells were present in almost all identified spatial domains across all developmental weeks, except the epithelium- and chondroblast-rich domains, where the highest mesenchymal cell content was below 4%. SOX2$^{high}$ epithelium-, SOX9$^{high}$ epithelium-, and chondroblast-rich domains were more homogeneous compared to other spatial domains, which were dominated by the corresponding individual cell types, constituting 59-89% of these domains (Fig. 4, Supplementary Fig. 15). This indicates the topographic segregation of these domains during the early formation of the organ. The most noticeable temporal change in the composition of the domains was that in 6-week-old lung, SOX2$^{high}$ epithelial cells and SOX9$^{high}$ epithelial cells constituted one single spatial domain, whereas from week 8.5 onwards, these cell types dominated their individual spatial domains.

Hierarchical clustering of correlations across all individual spatial domains and developmental weeks confirmed that domains enriched for the same cell type across different developmental ages exhibited greater similarity to each other than those enriched for different cell types within the same developmental age (Supplementary Fig. 16). Domains rich in airway smooth muscle cells, chondroblasts, and SOX2$^{high}$ or SOX9$^{high}$ epithelium clustered distinctly, indicating consistent spatial organization across different weeks of development. The more heterogeneous mesenchyme-, endothelium-, and immune-rich domains clustered together (Supplementary Fig. 16), reinforcing previously described patterns of cell type adjacency among these cell types. Overall, our data suggest that recurring organizational domains being to emerge in developing human lung tissue as early as week 6, laying the foundation for the mature lung tissue architecture.

## Proliferation patterns during early human lung development

We next explored the outgrowth patterns in the developing lung by studying the degree of proliferation in different cell types at different stages. To achieve this, we analyzed the Ki67 expression of the individual cell types as a proxy for their proliferation[34] (Fig. 5A, Supplementary Fig. 17). We observed a gradual decrease in proliferation over time, with 36% of all cells being proliferative in week 6, as compared to 14% at week 13 (Fig. 5B). Although proliferating cells were observed among all cell types, the majority of proliferating cells consisted of mesenchymal and endothelial cells (Fig. 5D).

In earlier weeks, SOX9$^{high}$ epithelial cells contained the highest fraction of proliferating cells (60–70%) compared to other cell types, followed by SOX2$^{high}$ epithelial cells. In later weeks, endothelial cells were the other most proliferating cell type whereas airway smooth muscle cells and lymphatic endothelial cells contained the lowest fraction of proliferating cells (Fig. 5C). In earlier weeks, there was a wide difference in the degree of proliferation across the different cell types, e.g. 73% in SOX9$^{high}$ epithelial cells versus 15% in airway smooth muscle cells at week 6. However, in later weeks the difference in the degree of proliferation was smaller across the different cell types, e.g. 17% in endothelial cells versus 7% in airway smooth muscle cells at week 13 (Fig. 5C). This suggests that the lung development is mainly focused

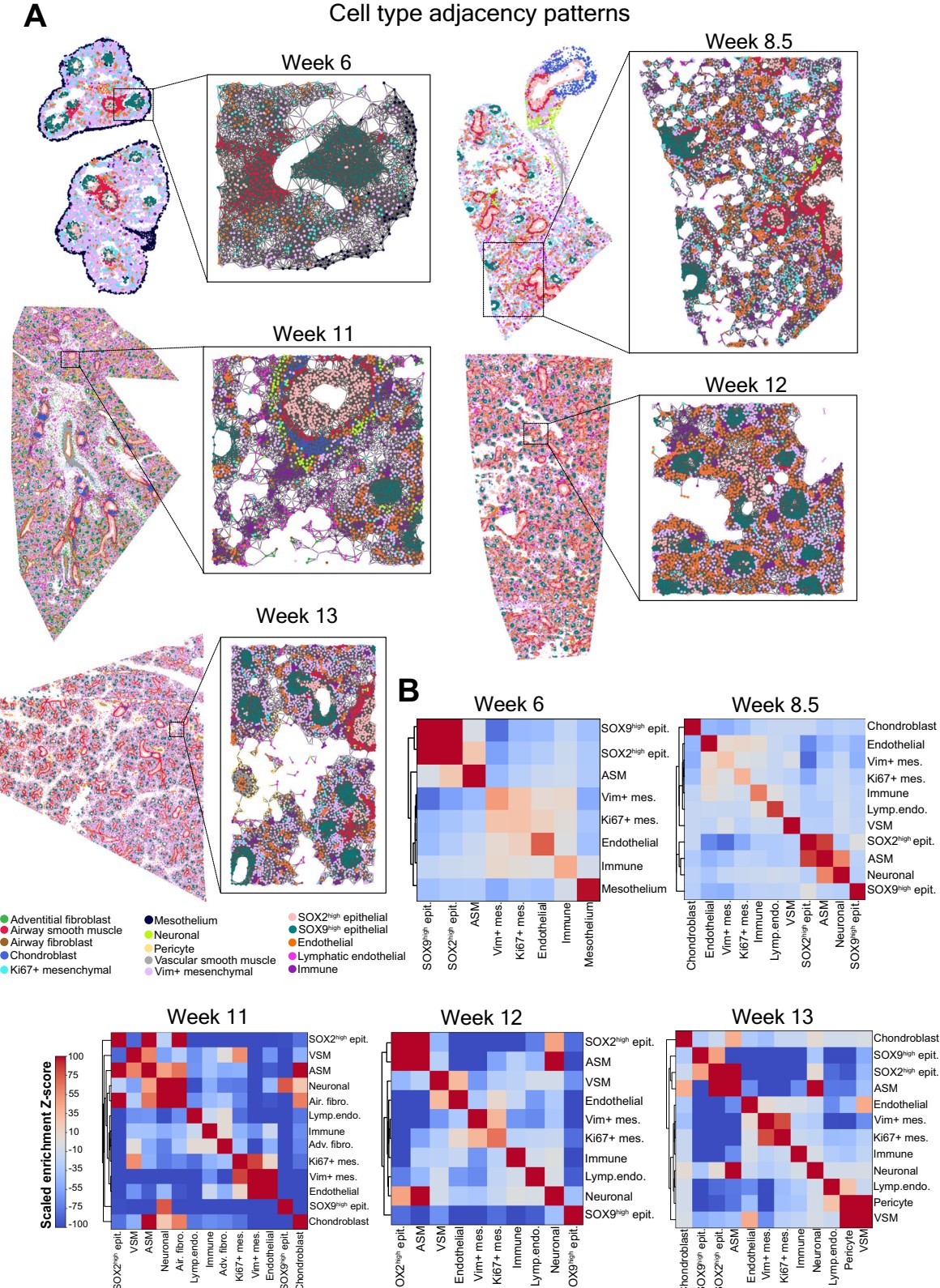

**Fig. 3 | Cell type adjacency patterns in the developing human lung tissue. A** Cell-cell adjacency graphs illustrating the entire tissue and a zoomed-in region. Each dot represents a single-cell colored according to its annotated cell type and physically neighboring cells are connected by edges. Cell type color assignments are displayed under the graph for Week 13. **B** Heatmaps with dendrograms summarizing the enrichment of cellular adjacency based on scaled Z-scores. Dark red color indicates cellular adjacency patterns with high enrichment Z-score and dark blue color indicates cellular adjacency patterns with low enrichment Z-score. ASM airway smooth muscle, VSM vascular smooth muscle, Adv. fibro Adventitial fibroblast, Air. fibro Airway fibroblast, Lymp. endo lymphatic endothelial.

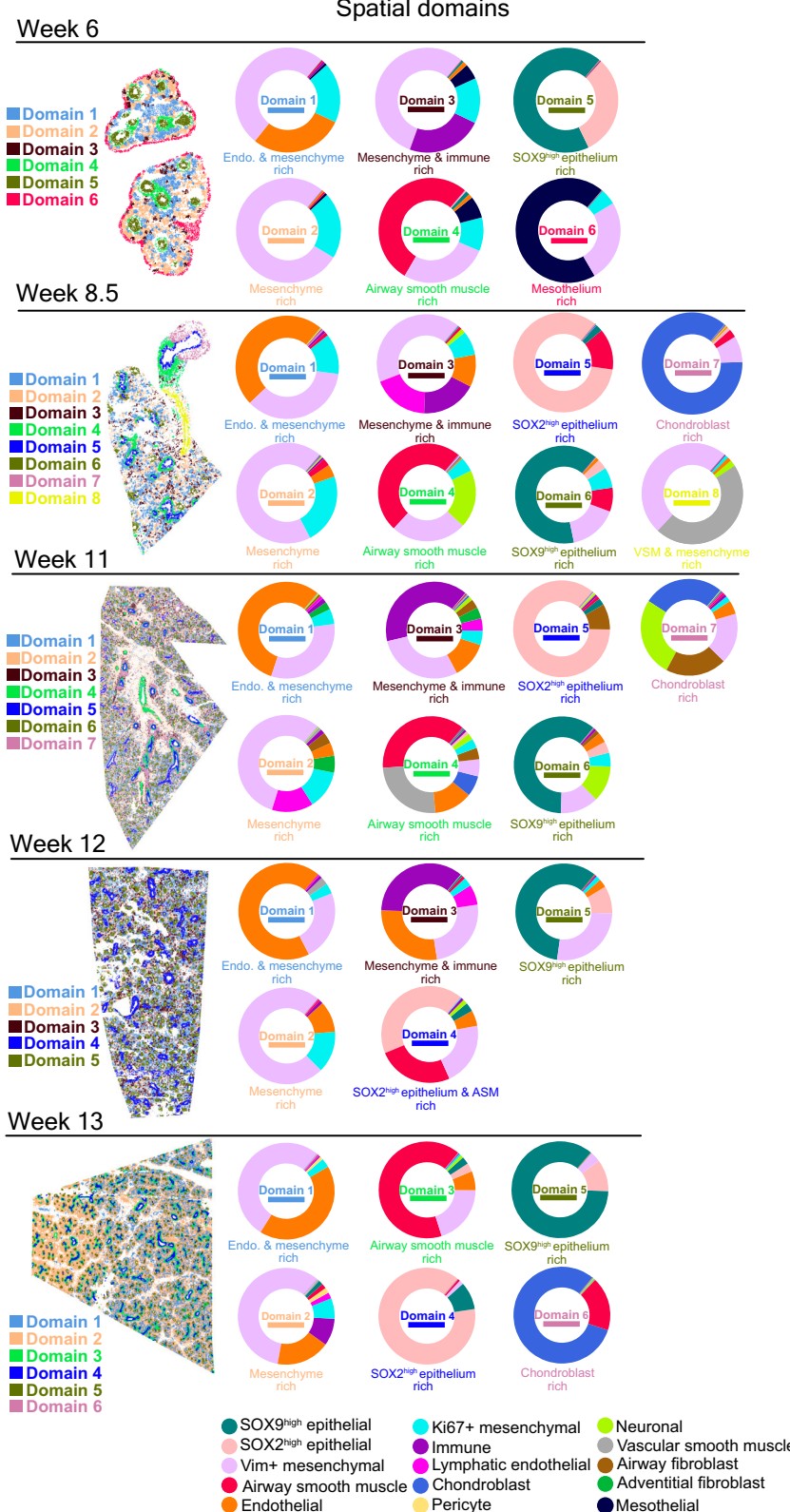

**Fig. 4 | Unsupervised identification of topographical structures in human developing lung tissue.** For each week, five to eight distinct spatial domains were identified and characterized by their cell type composition. Pie charts were used to represent these cell type compositions, employing the same color assignments consistent with previously annotated cell types. Cell type color assignments are displayed at the bottom. The identified spatial domains were visualized in spatial scatterplots, where domains with similar cell type composition across weeks were color-coded identically, albeit using a distinct color scheme from that of the annotated cell types. Source data for this figure is provided within the Source Data file.

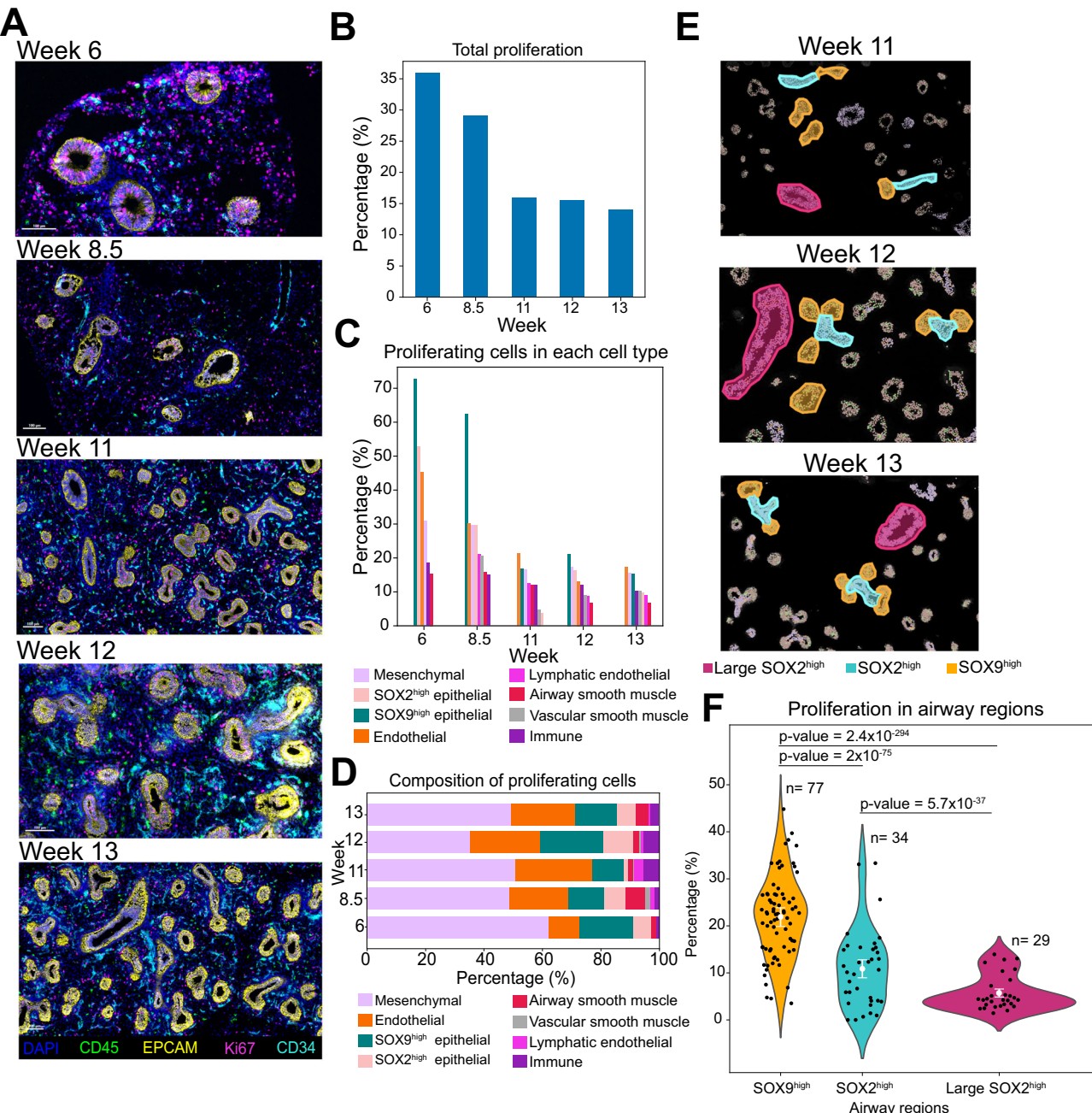

**Fig. 5 | Cell type specific proliferation patterns in the developing human lung tissue. A** Representative snapshots of Ki67+ proliferating cells in human lungs across different developmental weeks. DAPI in blue, CD34 (endothelial marker) in cyan, CD45 (immune marker) in green, EPCAM in yellow, and Ki67 (proliferation marker) in red. Images represent a single donor for each developmental week. Scale bar corresponds to 100 μm. **B** Bar plot displaying the fraction of proliferating cells among all cells analyzed for each developmental week. Source data for this figure is provided within the Source Data file. **C** Bar plot displaying the fraction of proliferating cells, separately, for each annotated cell type and developmental week. Cell type color assignments are displayed at the bottom. Source data for this figure is provided within the Source Data file. **D** Bar plot displaying the cell type distribution of proliferating cells for each developmental week. Cell type color assignments are displayed at the bottom. Source data for this figure is provided within the Source Data file. **E** Examples of SOX9high (orange), relatively small SOX2high (cyan) and relatively large SOX2high (magenta) epithelial airway structures in 11- to 13-week-old lungs, selected for analysis of proliferation patterns. **F** Violin plot displaying the distribution of the fraction of proliferating cells in in relatively smaller SOX2high, relatively larger SOX2high and SOX9high epithelial regions at weeks 11–13. The x-axis represents the different structures, and the y-axis indicates the percentage of proliferating cells in each of the (n) number of manually selected regions represented as a dot. Error bars represent mean ± 2×SEM (standard error of the mean). Statistical significance of the differences in degree of proliferation are summarized with a two-sided Fisher's exact test p-values. Source data for this figure is provided within the Source Data file.

on forming and expanding airways in the early weeks, and the endothelial and mesenchymal expansion, including smooth muscle, temporally follows the epithelial outgrowth.

Comparison of the enrichment of cellular adjacency among different proliferating cell types and their non-proliferating counterparts revealed that non-proliferating cells had generally more homotypic adjacency patterns compared to proliferating cells (Supplementary Fig. 18). A cumulative analysis across all weeks indicated that non-proliferating airway smooth muscle cells, immune cells, endothelial cells, SOX2high epithelial cells, and SOX9high epithelial cells, ordered by

statistical significance, exhibited significantly more homotypic adjacency patterns compared to their proliferating counterparts (Supplementary Fig. 18A). Conversely, mesenchymal cells showed the opposite trend, with proliferating mesenchymal cells displaying a higher degree of homotypic cellular adjacency pattern compared to their non-proliferating counterparts, which might imply the presence of centralized niches of mesenchymal proliferation during early human lung development.

We investigated further the proliferation patterns in airway structures in later weeks by comparing the degree of proliferation in SOX2[high] proximal airway regions and SOX9[high] distal tips (Fig. 5E, Supplementary Fig. 19A, B). Notably, the SOX9[high] distal tips contained a significantly higher fraction of proliferating cells compared to SOX2[high] proximal airway regions, which are found at the epithelial stalks (Fig. 5F). Furthermore, from week 12 onwards, the relatively smaller proximal airway regions contained a significantly higher fraction of proliferating cells compared to larger ones (Supplementary Fig. 19C). These results show that while the bulk of proliferation occurs in the distal tips, the cells in the stalks are more quiescent. This indicates a positional hierarchy of proliferation along the proximal-distal axis of the developing epithelial network. In addition, the proliferative cells in the large proximal airways presumably contribute to their diameter expansion. Collectively, these differential proliferation patterns in the epithelium provide valuable insights into the mechanisms of airway branching and elongation during early human lung development.

### Immune landscape in early human lung development

We next analyzed the immune compartment in depth by subclustering the immune cells (Fig. 6A, Supplementary Fig. 20). We observed that macrophages were the most abundant immune cell type during early lung development, supporting previously reported transcriptomics-based findings[28] (Fig. 6B). Additionally, our data revealed that macrophages became positive for the endocytic receptor MRC1 (CD206) between week 6 and 8.5, possibly pinpointing the period during which the phenotypic heterogeneity of macrophages starts to increase (Fig. 6A). B and NK cells were relatively less abundant immune cell types during early lung development. ILC & T cells emerged from 8.5 weeks onwards, and they were the second most abundant immune cell type except for week 12, where dendritic cells were the second most abundant cell type (Fig. 6B). Dendritic cells, NK cells and B cells emerged from 11 weeks onwards (Fig. 6B) emphasizing the distinct timelines of immune cell presence during development.

The immune cell type containing the highest fraction of proliferating cells was B cells (37–62% between weeks 11–13) and while the fraction of proliferating B cells increased, the fraction of proliferating macrophages and ILC & T cells decreased gradually (Fig. 6D).

Analysis of cell type adjacency patterns among the different immune cell types revealed that macrophages, ILC & T cells and dendritic cells were sharing proximity more with each other as compared to B cells and NK cells with more homotypic cellular adjacency patterns (Supplementary Fig. 21). As shown in Fig. 6E, macrophages were more frequently in proximity with ILC & T cells and dendritic cells, and less frequently with B cells and NK cells. These spatial proximity patterns especially between macrophages and ILC & T cells might have important implications for tissue homeostasis[35].

Starting at week 11, we observed groups of CD45[+] immune cells encircling the arterial vessels in the developing human lung tissue (Fig. 7A). We manually selected these artery-close immune cells in the images from weeks 11 to 13 (Supplementary Fig. 22A–C) and a comparison of their neighborhood composition within a radius of 50 μm distance to the neighborhood composition of randomly selected cells of any type confirmed their distinct spatial arrangement around the vasculature (Supplementary Fig. 22D). Comparison of their neighborhood composition to the one of randomly selected artery-distant immune cells in the rest of the tissue revealed that while the closest neighbors for both artery-close and randomly positioned artery-distant immune cells consisted predominantly of other immune cells, mesenchymal cells, and endothelial cells, the closest neighbors of artery-close immune cells were other artery-close immune cells, vascular smooth muscle cells, and pericytes (Fig. 7B). These artery-close immune cells were mostly macrophages and ILC & T cells (Fig. 7E). Differential marker analysis between the artery-close and artery-distant immune cells revealed that artery-close immune cells exhibited relatively higher levels of markers such as CD90 and HLA-DR, and relatively lower levels of CD163 and MRC1 (Fig. 7C). A complementary image analysis approach using intensity threshold-segmentation (Supplementary Fig. 23) confirmed these differential expression patterns (Fig. 7D). Concurrently, expression patterns for the corresponding genes, as well as for vascular remodeling factors such as VEGF-A within macrophage populations identified in single-cell transcriptomics data reported by Sountoulidis et al.[27] further supported the presence of distinct immune cell subpopulations potentially involved in vascular remodeling (Supplementary Fig. 24). Together, these findings suggest that the spatial arrangement of artery-close immune cells may be linked to their possible role in vascular growth and remodeling processes during human lung development.

## Discussion

Human lung development during the first trimester is a dynamic process marked by intricate morphological changes and interactions among different cell types. Our study delved into this complexity by analyzing around 1 million single cells in situ in human lungs aged 6 to 13 post-conception weeks using an in-house developed 30-plex antibody panel. We characterized the composition and spatial organization of cell types within these developing human lungs and tracked the temporal evolution and establishment of stable spatial domains representing functional tissue units. Additionally, we explored proliferation patterns across different cell types and microanatomic regions of the developing lung tissue.

Lung development hinges on key processes including cell proliferation, differentiation, migration and proximal-distal patterning. Our multiplexed antibody panel was meticulously curated to target markers involved in as many of these processes as possible. Rigorous testing and validation of over 70 antibodies resulted in an optimized 30-plex assay. Thus, our study not only presents a comprehensive proteomic view of early lung development but also serves as a resource documenting detailed performance of over 70 antibodies on fresh-frozen human developmental lung specimens.

The 30-plex antibody panel enabled us to localize the major types of cells present in the developing human lung. Comparison of cell type abundances derived from this protein-based spatial analysis to our previous single-cell transcriptomics analysis[27] revealed similar trends with notable differences in specific cell types such as endothelial cells, especially in week 12. These discrepancies may originate from a variety of factors, including tissue dissociation protocols causing a cell composition bias in single-cell transcriptomics data[36] or segmentation inaccuracies in our multiplexed protein imaging data. Crucially, these comparative analyses emphasize the distinct advantage of spatial omics techniques in capturing a relatively more faithful representation of cell type distribution within tissues.

One of the unique advantages of single-cell resolved high-parametric immunofluorescence is the possibility to explore the microenvironmental cell type configuration within the bona fide tissue space. This is particularly valuable in organs under development, as it enables a thorough understanding of the spatiotemporal dynamics governing organogenesis. Here, we observed that SOX2[high] proximal epithelial cells, SOX9[high] distal epithelial cells, and airway smooth muscle cells exhibited the most exclusive physical adjacency among themselves compared to other cell types. This robust interaction pattern suggests that these cells have the most densely interconnected

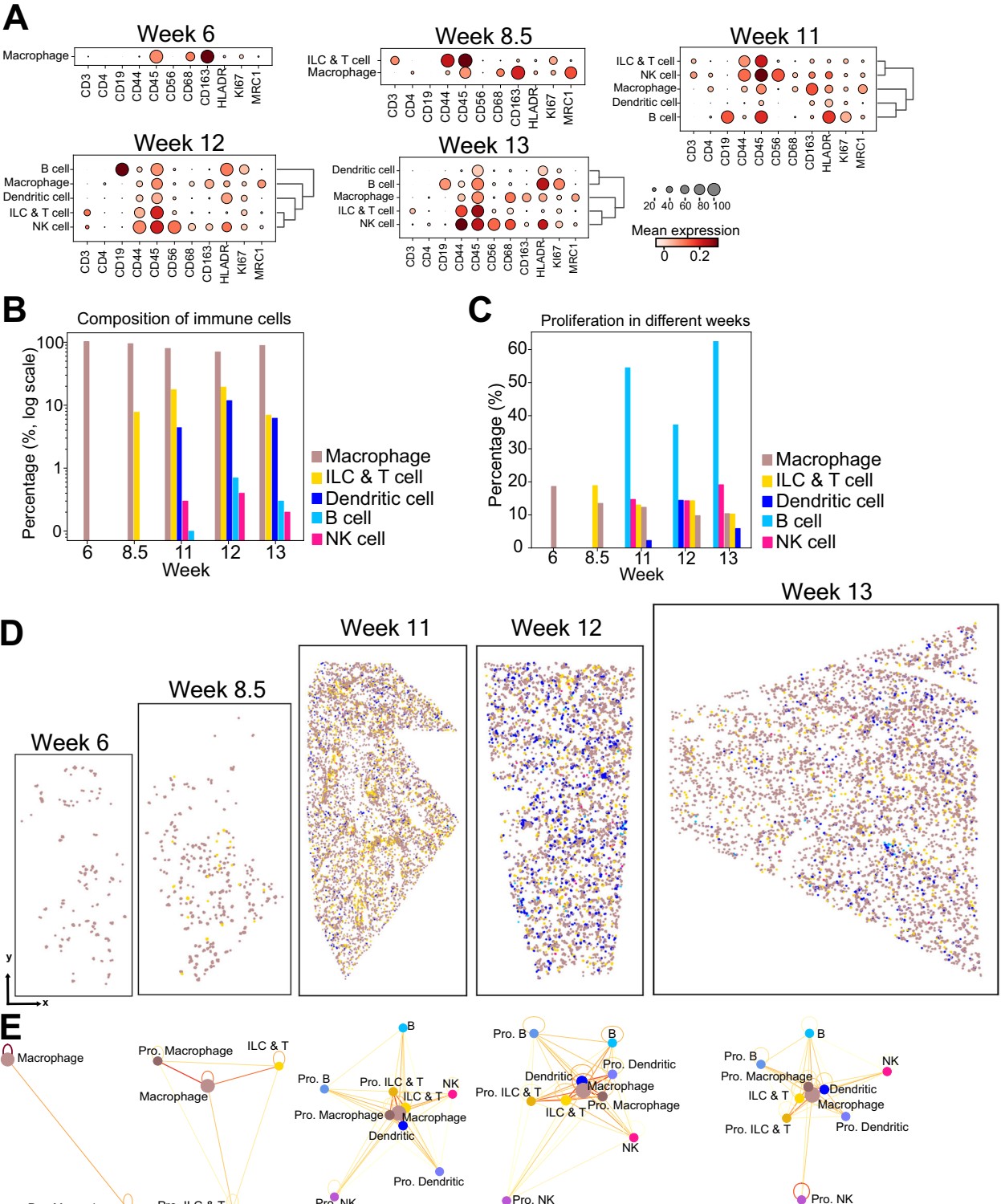

**Fig. 6 | Immune cell dynamics in developing human lung tissue. A** Dot plots illustrating marker expression in annotated immune cell sub-clusters across each developmental week. The size of the circles indicates the percentage of cells expressing the marker in each cluster. Dark red color indicates high and white color indicates low expression. **B** Bar plot displaying the composition of immune cell types, with the y-axis (log scale) indicating the percentage of each immune cell type and the x-axis representing the developmental week. Cell type color assignments are displayed to the right. Source data for this figure is provided within the Source Data file. **C** Bar plot displaying the percentage of proliferating cells in each immune cell type for each developmental week. Cell type color assignments are displayed to the right. Source data for this figure is provided within the Source Data file. **D** Spatial scatterplots summarizing the spatial distribution of identified immune cell types. Cell type color assignments are consistent with those in (**B**) and (**C**). **E** Network models summarizing the degree of physical proximity between the immune cell stypes including their proliferating (Pro.) subsets. The line color indicates the frequency of physical proximity between the cell types, where more frequent physical proximity between two different cell types is shown in red and less frequent physical proximity is shown in yellow. The color scheme for immune cell types is consistent with those in panels (**B–D**), with related shades used to represent their proliferating subsets.

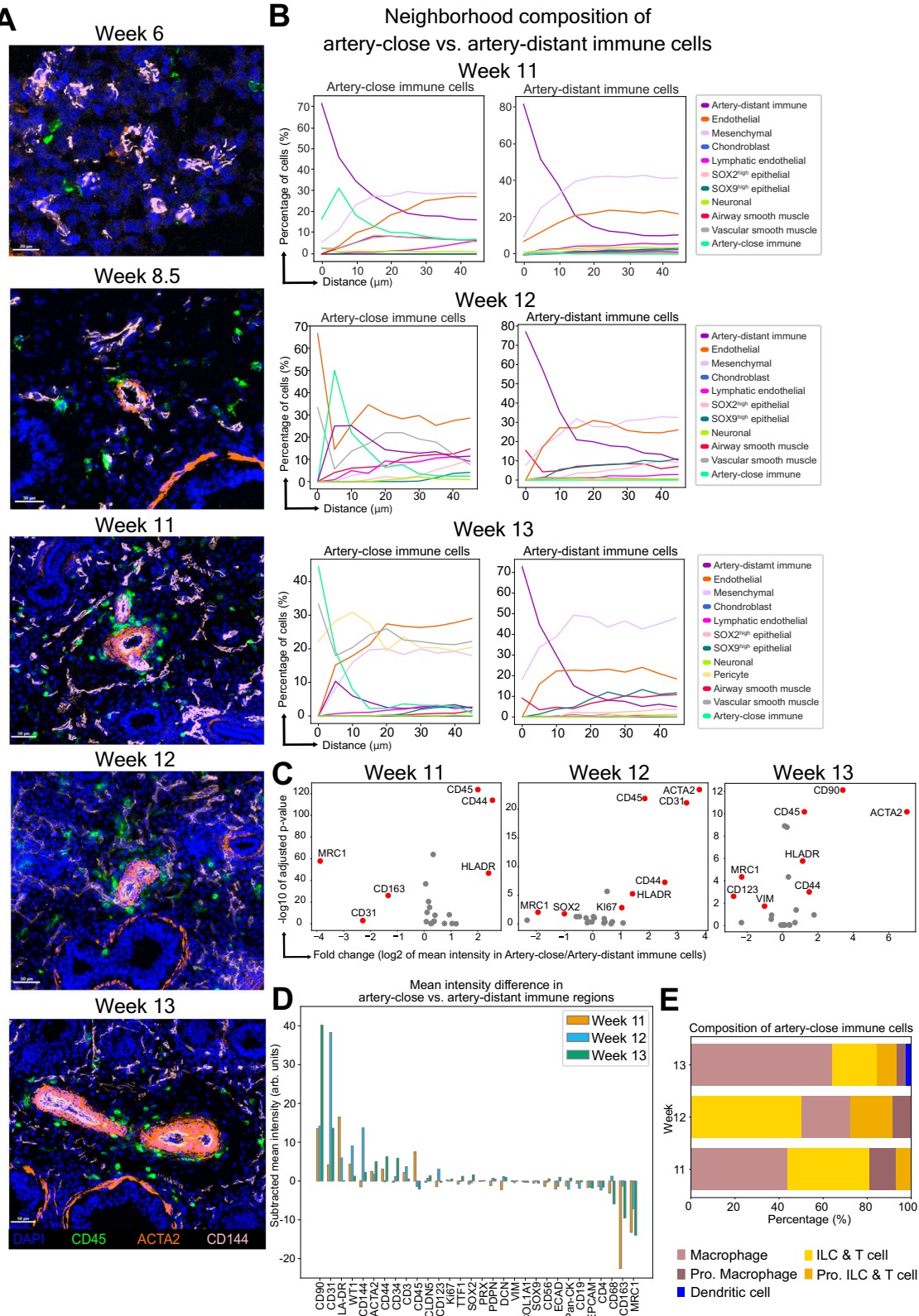

compartment compared to other cell types present within the mesenchymal tissue space, emphasizing the presence of distinct anatomical structures, and the presence of homotypic interactions in these distinct structures. Unsupervised analysis of spatial domains revealed a change in the composition of epithelial domains after week 6. Initially, SOX2$^{high}$ and SOX9$^{high}$ epithelial cells coalesced into a single epithelium domain in week 6. However, in later weeks, we observed

segregation where either SOX2$^{high}$ or SOX9$^{high}$ cells dominated distinct spatial domains. Overall, our analyses of cell type adjacency patterns and spatial domains revealed the drastic changes primarily in the epithelium between week 6 and 8.5, aligning with the dynamic airway branching and differentiation processes occurring during the pseudoglandular stage, and underscoring also how the complex architecture is maintained from week 8.5 onwards.

**Fig. 7 | Immune cells encircling vasculature during later developmental weeks. A** Representative snapshots of immune cells and their spatial arrangement around arteries at different developmental weeks. DAPI in blue, CD45 (immune marker) in green, ACTA2 (vascular smooth muscle marker) in orange and CD144 (endothelial marker) in pink. Images represent a single donor for each developmental week. Scale bar corresponds to 20 μm for week 6, 30 μm for week 8.5 and 50 μm for weeks 11 to 13. **B** Line plots summarizing the neighborhood composition of artery-close (left) and artery-distant immune cells (right) within a radius of 50 μm for weeks 11–13. The x-axis represents the distance in μm, and the y-axis represents the percentage of neighboring cells. Cell type color assignments are displayed to the right. Source data for this figure is provided within the Source Data file. **C** Volcano plots summarizing differential marker expression between artery-close and artery-distant immune cells for weeks 11–13 based on nucleus-segmented data. The -log10

of Benjamini-Hochberg adjusted two-sided Wilcoxon rank-sum test $p$ values plotted against the log2 of fold change for each marker. Markers with a $p$ value < 0.05 and an absolute log2 fold change >1.0 are indicated in red. **D** Differential marker expression analysis between artery-close and artery-distant immune cell regions for weeks 11 (orange), 12 (blue) and 13 (green), independent from single-cell segmentation and based on intensity threshold-based semantic segmentation for the markers ACTA2, CD144 and CD45 (details in "Methods" and Supplementary Fig. 23). The bar plot summarizes the ordered intensity difference for each marker, where the mean intensity in arbitrary (arb.) units in artery-distant immune cell regions was subtracted from artery-close immune cell regions. Source data for this figure is provided within the Source Data file. **E** Composition of immune cell types encircling arteries. Cell type color assignments are displayed at the bottom. Source data for this figure is provided within the Source Data file.

The developing lung undergoes dramatic tissue growth and remodeling, during which proliferation and apoptosis occurs[37]. Investigating the degree and change of proliferation patterns across different cell types revealed a gradual decrease in proliferation. SOX2[high] and SOX9[high] epithelial cells were the cell types with the highest fraction of proliferating cells in week 6 and their degree of proliferation decreased over time indicating an extensive branching in the airways in earlier stages of development, confirming a similar pattern in proliferation reported for slightly later stages of lung development[33]. In animal models, hyperproliferation of epithelial and mesenchymal cells during fetal development is reported to disrupt distal airway formation and mesenchymal and vascular remodeling[38]. Compromised proliferation is also indicated in the context of rare congenital anomalies such as pulmonary hypoplasia[39]. Thus, our protein level data reporting the degree of proliferation provides a valuable resource to benchmark proliferation patterns in the developing human lung.

Our data revealed that SOX9[high] distal tips contain a higher fraction of proliferating cells as compared to SOX2[high] proximal regions, confirming that the lung epithelial network expands mainly through the outgrowth of the distal SOX9[high] epithelial cells. Our data also showed that the degree of proliferation in smaller SOX2[high] proximal airways is higher than in larger proximal airways, suggesting that proliferation at these stalk regions locally contributes to the branch elongation. Collectively, epithelial proliferation followed a graded pattern along the distal to proximal axis of the organ. Following epithelial cells, endothelial cells were the second most proliferative cell type among others. This finding suggests a synchronized progression of airway branching with vascularization, supporting previous research[40].

Immune cells do not only play an important role in lung homeostasis[2] but they also contribute to organ development. Here we show how immune cells get diversified over a span of only seven weeks and how they gradually build a spatial network to cover the entire tissue. We identified macrophages as the most abundant immune cell type in early developmental human lung. Interestingly, our study pinpointed also that human lung macrophages become MRC1[+] (CD206[+]) between post conceptional week 6 and 8.5 and complements previous studies reporting that human neonates are born with MRC1[+] lung macrophages[41,42]. Compared to macrophages and ILC & T cells, NK and B cells were less abundant during the developmental period of 6 to 13 weeks. Previous transcriptomics-based studies reported that these cells become prominent first after week 15[28]. Here, we show the presence of NK and B-cells in week 11 onwards using immunofluorescence. A more recent study[29] reported the presence of dendritic cells from week 8 onwards using the marker CD1C for immunofluorescence. Although our panel did not include such an exclusive marker for dendritic cells, we were able to annotate them in later weeks based on both HLA-DR expression and the absence of other lineage markers. Thus, we here report the presence of dendritic cells from

week 11 onwards, while acknowledging that they may also be present in earlier weeks. Our data also revealed that while the fraction of proliferating macrophages decreased during the investigated period, the fraction of proliferating B cells increased, and B cells contained the highest fraction of proliferating cells. This aligns with previous reports stating that although B cells are rare, they express proliferation markers[29,43].

Our study also revealed the presence of immune cells, predominantly macrophages, in a ring-shaped arrangement encircling the developing vessels. This was previously described in prenatal murine lung and was observed to disappear right after birth[43]. This transient accumulation of lung macrophages around the developing vessels, temporally restricted only to prenatal development, suggests their function in modulating vascular growth and remodeling. Our data implies the same possible function for macrophages in the developing human lung. In addition to macrophages, we also identified the presence of CD3 expressing cells in these regions, indicating a possibly broader spectrum of immune cell types than previously recognized. Previous studies, not on prenatal but adult murine lungs[44], reported the presence of group 2 innate lymphoid cells (ILC2s) encircling arteries. While our panel did not allow us to distinguish ILC2 cells from other ILC and T cells, our observations align with the literature[43,44], suggesting a potential involvement of also these cells in modulating vascular growth and remodeling in developing human lung.

Our study has a few limitations. First, although our 30-plex antibody panel successfully identified the main cell type compositions in epithelial, mesenchymal, and immune compartments, it offered limited depth of analysis for certain key cell types such as secretory or neuroendocrine cells. Second, the reported data is based on the analysis of lung samples from embryos/fetuses aged 5-13 post-conception weeks, sourced from one donor at each developmental time point, covering five timepoints in total. Analyzing more timepoints would provide a higher resolution of the temporal changes in lung development. Although no malformations were observed in any of these embryos, and our protein-based results were consistent with transcriptomics data across multiple embryos[27]—making it unlikely that our findings reflect abnormal lung development due to chromosomal defects—another limitation is that these embryos were not karyotyped. Finally, the lung tissue sections from different developmental weeks did not offer an equally balanced representation of the distal or peripheral microanatomy, leading to inconsistent identification of certain cell types, such as chondroblasts and adventitial fibroblasts, in our downstream analysis. Larger, more uniform, and independent cohorts of tissue sections are needed to yield sufficient statistical power to confirm the protein expression and relationship patterns, as well as their temporal changes, described in our study.

Given the lack of prior published reports using a comprehensive multiplexed antibody-based approach for spatial analysis of human embryonic lung tissue, this study uniquely provides a single-cell resolved protein-level expression resource to decode the composition

and spatial relationships of main cell types present in the human embryonic lung during the first trimester. It provides important insights into the spatial diversity of developing human lung and serves as a valuable resource that the pulmonary and developmental biology community can explore to further understand the human lung formation and function at protein-level. The dataset, now openly accessible for in-depth exploration via an interactive portal (https://hdca-sweden.scilifelab.se/tissues-overview/lung/), provides a new avenue to unravel the developmental origins of health and disease in the human lung.

## Methods

### Ethics statement and prenatal tissue samples

The use of human prenatal material from the elective routine abortions was approved by the Swedish National Board of Health and Welfare (Socialstyrelsen). The molecular analysis using this material was approved by the Swedish Ethical Review Authority (Etikprövningsmyndigheten) (2018/769-31).

The lung samples from embryos/fetuses aged 6–13 post-conception weeks (pcw) were sourced from one donor at each developmental time point. The tissue donors were recruited among pregnant females seeking elective abortions. Referral to hospitals was managed by a central office for all abortion clinics in the Stockholm region. The recruitments were conducted by midwives uninvolved in the conducted research, ensuring no bias in the selection of participants. Both oral and written consent were acquired from the tissue donors that the retrieved embryonic/fetal material can be used for research purposes and that they are able to withdraw their consent at any time. No compensation of any kind was provided to the tissue donors. Inclusion criteria included donors being 18 years of age or older and fluent in Swedish. Exclusion criteria excluded abortions performed for medical reasons, socially compromised donors, or any indications of uninformed consent. The donors were self-reported as "healthy".

No major malformations were observed in any of the embryos used in the study. At time of collection of embryonic/fetal tissues, the sex of the embryos/fetuses was not determined, thus no biased selection was performed. The sex of the 8.5- and 13-pcw-old embryos was subsequently assigned as female and the sex of the 11-pcw-old embryo was assigned as male based on *XIST* expression[27]. Information regarding the sex of the 6-pcw and 12-pcw-old embryos was not available. The age of each embryo/fetus was determined using clinical ultrasound, actual crown-rump length and other anatomical landmarks.

### Tissue treatment for spatial analyses

The left lung from each donor was snap-frozen in cryo-matrix for spatial analyses. The sectioning of 10 µm-thick tissues was performed with a cryostat (Leica CM3050S) on slides (Epredia, J1800AMNZ) as previously described[27]. Sections were dried on a silica gel at 37 °C for 15 min and then stored at −80 °C.

### Immunostaining for high-parametric imaging

After obtaining the tissue sections from −80 °C, samples were put on Drierite beads (Alfa Aesar, K13Z026) and the staining was performed according to the manufacturer's protocols (Akoya Biosciences, Marlborough, MA). Tissues were first incubated in acetone (Sigma, 650501) for 10 min, then in hydration buffer (Akoya Biosciences, 7000008) for 2 min, and fixed in 1.6% paraformaldehyde (Thermo Scientific, 043368.9 M) diluted in hydration buffer for 10 min. Tissues were then incubated in a staining buffer (Akoya Biosciences, 7000008) for 30 min. After equilibration, tissues were incubated with blocking buffer (Akoya Biosciences, 7000008) and primary antibody cocktail (dilutions indicated in Supplementary Table 1) for 3 h inside a humidity chamber and at room temperature (RT). The samples were washed

with a staining buffer for 2 min, and then fixed in 1.6% PFA diluted in a storage buffer (Akoya Biosciences, 7000008) for 10 min. The samples were then washed with phosphate-buffered saline (PBS), incubated in cold methanol (Sigma, 322415) (4 °C) for 5 min, and washed again with PBS. Finally, the samples were fixed in a fixative reagent (Akoya Biosciences, 7000008) and washed again with PBS before being stored in storage buffer at 4 °C.

### Indirect immunofluorescence

Samples were equilibrated at RT for 15 min and fixed with 4% paraformaldehyde (Thermo Scientific, 043368.9 M) for 15 min. After fixation, samples were washed 3x in PBS for 15 min. Antibodies, diluted in PBS containing 0.3% Triton (Sigma, T8787), were added (200 µl /slide) and samples were incubated overnight at 4 °C. The samples were blocked for 30 min in 1xTBS (Medicago, 097500100) containing 0.5% TSA blocking reagent (Perkin Elmer, FP1020) and Hoechst (Invitrogen, H3570). Secondary antibodies were diluted in a blocking buffer and incubated for 90 min at RT. After incubation, slides were washed 3x with TBS-Tween for 15 min. Coverslips (VWR, 631-0147) were mounted on the slides using Fluoroshield mounting medium (Invitrogen, 00495802).

### Antibody conjugation to build multiplexed panels

The high-parametric antibody panel consisted of both a selection of pre-conjugated antibodies ($n = 23$, Akoya Biosciences) and antibodies available through other commercial sources that were conjugated in house ($n = 49$, Supplementary Table 1). The latter were free from additives or preservatives that might potentially affect the conjugation efficiency. For conjugation, the following protocol was used (Akoya Biosciences): Filters with 50 kDa molecular weight cut off were first blocked using 500 µl of filter blocking solution (Akoya Biosciences, 7000009). For each conjugation, 50 µg of antibody was used and if the volume was less than 100 µl, PBS was added to reach 100 µl. Antibodies were transferred on blocked filters and centrifuged at $12,000 \times g$ for 8 min. The reduction mixture (Akoya Biosciences, 7000009) was prepared, put on the filters containing antibodies, and incubated for 30 min at RT. The reduction solution was removed by centrifugation ($12,000 \times g$) and exchanged with the conjugation solution (Akoya Biosciences, 7000009). Each oligonucleotide barcode was rehydrated in 10 µl nuclease-free water (Life Technologies, AM9937) and 210 µl of conjugation solution was added. The corresponding barcodes were added on top of the reduced primary antibodies and incubated for 2 h at RT. After centrifugation ($12,000 \times g$), purification solution (Akoya Biosciences, 7000009) was added, and antibodies were purified in several steps of centrifugation. Then, 100 µl of antibody storage buffer (Akoya Biosciences, 7000009) was added on top of antibodies and centrifuged ($3000 \times g$) for 2 min.

Antibody conjugation was validated by manual incubation with PhenoCycler reporters: After staining with the conjugated antibodies, the tissue was incubated in a screening buffer containing 10x PhenoCycler buffer (Akoya Biosciences, 7000001), nuclease-free water and DMSO. Meanwhile, a reporter stock solution was prepared including screening buffer, assay reagent (Akoya Biosciences, 7000002) and nuclear stain (Akoya Biosciences, 7000003). Here, 2.5 µl from each reporter was mixed with the reporter stock solution and the end volume was 100 µl for each slide. Slides were incubated with the reporters for 5 min in the dark and then washed with the screening buffer several times before mounting on a slide using Fluoroshield mounting medium (Invitrogen, 00495802).

### Automated collection of high-parametric immunofluorescence data

Reporters were diluted in reporter stock solution composed of nuclease-free water (Life Technologies, AM9937), 10x PhenoCycler buffer, assay reagent, and nuclear stain. Diluted reporters were put

into the corresponding wells according to the experimental plan. Fully automated staining and imaging was performed using the PhenoCycler Fusion system (Akoya Biosciences) (Supplementary Fig. 1). Raw 16-bit images were generated for each cycle using ×20 magnification (resolution: 0.51 μm/pixel). Acquired images were processed by the PhenoImager Fusion software (Akoya Biosciences, v 1.0.3) and 8-bit ".qptiff"-image files were obtained as output for subsequent analyses.

## Analysis of high-parametric imaging data

**Image processing and segmentation.** The ".qptiff" files were opened in FIJI[45] and were converted into separate.tiff images for each marker channel. These single-channel images were then uploaded to our in-house developed software PIPEX to perform instance segmentation and various downstream single-cell data analysis. For this, DAPI and EPCAM marker images were preprocessed as follows to identify the most significant signal intensity thresholds: First, background subtraction was performed by subtracting the lower 0.01 percentile from the signal intensity values. This was followed by a min-max normalization, where the minimum value was transformed into a 0 and the maximum value was transformed into a 1, and every other value was transformed into a decimal between 0 and 1. After this, Otsu's method was used for thresholding by implementing the skimage.filters.threshold_multiotsu function in the scikit-image package[46] using nbins=3 parameter.

The IF Stardist ML model[47] was executed over the preprocessed DAPI marker to get the pure and expanded nuclei shapes (nuclei masks). A watershed algorithm was executed over the preprocessed EPCAM marker to get the membrane boundary shape information (membrane mask). The nuclei mask expansions (not the pure nuclei shapes) were cut to fit the membrane boundaries calculated in the membrane mask. In the locations where no/incorrect nuclei were detected by Stardist, the membrane shapes themselves were kept as segmented cells. These were considered as final segmented cells. A sanity filtering was applied using cell size and a combination of DAPI/EPCAM intensities to discard segmented cells that were too small and/or inconsistent with the overall intensity levels. With this final combined mask,.csv files were generated for all segmented cells, including mean intensities and binarizations for all the original marker images (including original DAPI and EPCAM images) (Supplementary Fig. 3).

Next, Perseus[48] was used for each sample to exclude the areas presumably not belonging to the lung tissue, or showing sectioning or staining artifacts (Supplementary Fig. 4A). The output after this data clean-up was used for the downstream quantitative analysis. A manual background subtraction was performed prior to the clustering step. For each marker, images were inspected using FIJI, where background values were measured in regions of interest (ROIs) where no signal was observed and where the background signal appeared relatively high (Supplementary Table 2). Background values defined for each marker were subtracted from the values in the segmented dataset.

**Clustering and annotation of cell populations.** Segmented and background-subtracted values were used to generate annotated data matrices using Anndata[49]. Data was then normalized using Arcsinh normalization[50]. To identify the cell type populations present in situ, Scanpy's[51] PCA (sc.tl.pca(adata)), find neighbors (sc.pp.neighbors(adata,n_neighbors = 10)), UMAP (sc.tl.umap(adata,min_dist = 0.1, spread = 0.3, negative_sample_rate = 4)) and Leiden clustering (sc.tl.leiden(adata,resolution = 1)) functions were used. After the first round of clustering, clusters were investigated and artifact clusters were removed (Supplementary Fig. 5, Supplementary Fig. 6).

Datasets of each sample after artifact removal were merged using ComBat[52] (sc.pp.combat(adata, key = 'age')). The same parameters as for the initial clustering were used for clustering of the merged data. Density plots (sc.pl.embedding_density(adata, groupby = 'age') were used to investigate the efficiency of data merging (Supplementary

Fig. 7A). The merged data was visualized in a UMAP colored for each week (Supplementary Fig. 8A) and clusters of the merged dataset were annotated for each cell type in accordance with the dot plots (sc.pl.dotplot (adata, markers, groupby= 'leiden', dendrogram=True, size_title=None) (Supplementary Fig. 8B). Based on the cell type annotations, the percentage of each cell type was calculated (Supplementary Fig. 8C).

For further analysis of individual weeks, clustering was performed for each week separately using the previously mentioned parameters. Cell type annotations were performed based on the dot plots (Supplementary Fig. 9). After annotations, clusters belonging to the same cell type were merged (Supplementary Fig. 10). Some cell types were annotated further for positivity for certain markers such as Vim+ (Vimentin positive) mesenchymal or Ki67+ (Ki67 positive) mesenchymal. The airway smooth muscle clusters were sub-clustered enabling identification and annotation of neuronal cell clusters for weeks 12 and 13 for the downstream analysis. Here, the parameters were the same as the parameters previously mentioned, except for sc.tl.leiden function a resolution of 0.5 was used.

For calculating and comparing the main cell type percentages in each week, clusters with detailed annotations were merged back with their main cell type cluster, e.g. Vim+ mesenchymal, airway fibroblast, adventitial fibroblast, neuronal, pericyte and chondroblast clusters were combined as "mesenchymal cluster" for comparative analyses across different weeks. Correlation between the expression patterns of markers was calculated using Pearson's correlation (Supplementary Fig. 13).

A comparison of cell type annotations for each cell when clustered using either the merged or individual timepoint data revealed that while a majority (78%) of the cell type assignments were consistent, discrepancies existed in the assignment of certain cell types, such as SOX2[high] epithelial cells, immune cells and airway smooth muscle cells (Supplementary Fig. 11). Further examination of these discrepancies revealed that the cell type assignments based on individual timepoint data were more accurate than those based on the merged data. Consequently, all in-depth downstream analyses were conducted using datasets from individual timepoints rather than the merged data.

In order to do a comparison between cell type percentages obtained in our current protein and previous transcriptomics datasets, single-cell RNA sequencing data from Sountoulidis et al. [27] was used and cells were grouped into similar cell type annotations (Fig. 2C).

**Analysis of cell type adjacency.** Spatial localization patterns of annotated cell type clusters were analyzed using spatial neighbors function (sq.gr.spatial_neighbors(adata, radius = 50, coord_type = "generic")) in Squidpy[53], which creates a graph from spatial coordinates based on a selected fixed radius, which was 50 for this analysis. To visualize the connectivity between cells, scatter plots were used (Fig. 3A). To quantitatively analyze the cell type adjacency patterns, enrichment analysis was performed (sq.gr.nhood_enrichment(adata_new, cluster_key = "annotation")), and enrichment Z-scores were plotted in heatmaps (sq.pl.nhood_enrichment(adata_new, cluster_key=annotation, method = "ward", mode = 'zscore', vmax = 100,vmin = −100, cmap = 'coolwarm', show=False) (Fig. 3B). Next, the Z-scores for each week was scaled using scale(data_frame, center = True, scale = True) function in R. For further comparison across different weeks, only the common cell type annotations were kept in the scaled dataframes such as airway smooth muscle, endothelial, immune, Ki67+ mesenchymal, SOX2[high] epithelial, SOX9[high] epithelial and Vim+ mesenchymal cell types. Then, the scaled enrichment Z-score dataframes for all weeks were combined using df.to_numpy() function and stacked with np.stack(data_matrices, axis = −1) function to create a 3D dataframe. To visualize the changes in cell type adjacency patterns over time, time-series plots were generated for each cell type displaying its adjacency to other cell types. For each cell

type, a separate line plot was created based on the scaled enrichment Z-scores across the different weeks, where x-axis represents the weeks and y-axis the scaled Z-score values (Supplementary Fig. 14).

**Identification of spatial domains.** For unsupervised analysis of the tissue architecture by identification of tissue domains, BANKSY algorithm[54] was used based on its comparative performance against similar algorithms[55]. Tissue domains were independently identified in each developmental week by using mixing parameter lambda = 0.8. After this, domains were annotated based on their cellular composition (Fig. 4). Cell type composition within each domain was calculated as percentages (Supplementary Fig. 15) and similarity between cell type compositions in spatial domains across different weeks was assessed by calculating the Pearson's correlation coefficient (Supplementary Fig. 16).

**Analysis of proliferation patterns.** To determine proliferation patterns, we used the background-subtracted data frame for all weeks. For each individual week, we determined a Ki67 threshold value that separated the cell population of that week into Ki67-positive and -negative cells based on the image data. The thresholds corresponded to the 85th percentile for weeks 11 and 12, the 70th percentile for weeks 6 and 8.5, and the 86th percentile for week 13. Based on these intensity thresholds, Ki67+ cells in the Anndata object were annotated as 'proliferating'. Proliferating cells were counted, and proliferating fractions were calculated for each week (Fig. 5B–D). To compare the degree of proliferation across different cell types in different weeks, broadly annotated cell types were used (Supplementary Fig. 18A). Spatial localization patterns of proliferating and non-proliferating cells were analyzed using sq.gr.nhood_enrichment(adata, cluster_key = "proliferation") function to generate Z-score outputs. Similar to the 'analysis of cell type adjacency' section, the Z-score dataframes for each week were scaled using scale(data_frame, center = True, scale = True) function in R. As described previously, 3D dataframe was created using df.to_numpy() function together with np.stack(data_matrices, axis = −1) function. To compare the diagonal interactions, in other words, the cell-cell adjacency with itself; box plots with scaled Z-scores of non-proliferating and proliferating cell adjacencies were plotted and statistically compared using two-sample t-test for each cell type. Also, the p-values were calculated using the ttest_ind() function (Supplementary Fig. 18A). To provide an overview of the changes in cell-cell adjacency over time, the scaled Z-score enrichments for non-proliferating cell types were divided by the corresponding scaled Z-score enrichments for proliferating cell types. For example, the scaled Z-score enrichment for the adjacency of non-proliferating airway smooth muscle cells with themselves was divided by the scaled Z-score enrichment for the adjacency of proliferating airway smooth muscle cells with themselves. These ratios were plotted as a line plot across different time points (Supplementary Fig. 18B).

The degree of proliferation was compared further between SOX2high airway regions, larger SOX2high airway regions and SOX9high airway regions (Fig. 5E). For this,.h5ad files were uploaded to TissUUmaps[56] and airway regions were selected on images for each of these three categories (Supplementary Fig. 19B). Here, only 11-, 12- and 13-week-old samples were used, as the overall size of airway-like structures in earlier weeks did not allow to distinguish well further between these three structures. In these three categories of selected regions, the numbers of proliferating cells were calculated. Cell number counts for proliferating and non-proliferating cells were then used to perform a two-sided Fisher's exact test to calculate the statistical significance of differences in proliferation patterns in these regions (Fig. 5F, Supplementary Fig. 19C).

**Subclustering of immune cells.** For each week, the immune cell cluster was subclustered using the following markers: CD3, CD4, CD44, CD45, CD56, CD68, CD163, HLA-DR, Ki67 and MRC1. For clustering,

Scanpy's PCA (sc.tl.pca(adata)), find neighbors (sc.pp.neighbors(adata,n_neighbors= 10)), UMAP (sc.tl.umap(adata,min_dist =0.1, spread = 0.3, negative_sample_rate = 4)) and Leiden clustering (sc.tl.leiden(adata,resolution = 0.2)) was used. For week 8.5 and 12, a resolution parameter of 0.3 was used. The resulting clusters were annotated by inspecting their expression profiles in dot plots (Supplementary Fig. 20) and clusters with the same annotations were merged (Fig. 6A). In weeks 11,12, and 13, B and NK cell annotations were integrated to the analysis with the help of manual selection based on the marker expression and co-expression of CD19 and CD45 for B cells, and CD56 and CD45 for NK cells. The percentages of the cell types were calculated based on the cell counts (Fig. 6B). Based on Ki67 expression, proliferating immune cell subtypes were identified and integrated to the immune cell Anndata object. The fractions of proliferating cell types were calculated based on the cell counts (Fig. 6D). As described under "Analysis of cell type adjacency" section, spatial proximity between immune cell types and all the other cell types were calculated using sq.gr.spatial_neighbors with radius set to 50 and sq.gr.nhood_enrichment functions. The resulting enrichment Z-scores were plotted in heatmaps (Supplementary Fig. 21). A model of physical proximity between cell types was created by using the "Draw network" function of the UTAG package[57] with default settings and as an input 'count' from sq.gr.nhood_enrichment was used (Fig. 6E).

**Comparative analyses of immune cells encircling vasculature.** Groups of immune cells encircling arteries, hereinafter referred to as artery-close immune cells, were observed in 11-, 12- and 13-week-old samples. Comparative analyses for these cells' marker expression patterns and neighborhood compositions were performed using the so far described nucleus-based instance-segmented (i.e. single cell-segmented), and alternatively also threshold-based semantic-segmented imaging data.

**Comparative analyses using nucleus-based instance-segmented imaging data.** The immediate cellular neighborhood composition of the "artery-close" immune cells were compared both to randomly selected other, i.e. "artery-distant" immune cells, as well as to cells of any type, matching in both cases the artery-close immune cells' quantity. First,.qptiff files for these three weeks were loaded to QuPath[58] and with the help from endothelial cell marker CD144 and smooth muscle marker ACTA2, arterial structures in each tissue section were identified on the images. Immune cells encircling these arterial structures were selected manually based on the CD45 marker expression. These manually selected immune cells were collected in a.json file and cell ID information was gathered. Gathered cell IDs were added to the Anndata object and tagged as "arterial". To analyze the composition of cell types that were within a 50 μm radius around the selected artery-close immune cells, an in-house script based on calculation of Euclidean distance between the centers of two cells was used (Supplementary Fig. 22A, B). The same analysis was performed for the equal number of artery-distant immune cells (Fig. 7B), and also equal number of cells of any type (Supplementary Fig. 22D), by randomly selecting them using sample() function.

In addition to the comparison of their neighborhood composition, average marker expression patterns of all artery-close and all artery-distant immune cells were compared. For this differential expression analysis between artery-close and -distant immune cells, a two-sided Wilcoxon rank-sum test was applied. The sc.tl.rank_genes_groups function was used for the calculation (sc.tl.rank_genes_groups(adata_new, 'arterialimm', groups = ['Arterial Immune'],reference = 'Immune',method = 'wilcoxon',n_genes=adata_new.var_names.size,corr_method = 'benjamini-hochberg'). Markers with an adjusted p-value < 0.05 and an absolute log2 fold change > 1.0 were considered significant. For visualization, a volcano plot was created, where -log10 of the adjusted p-values were plotted against

the log2 fold changes. Significant markers were highlighted in red and annotated in the plot (Fig. 7C).

**Comparative analyses using intensity threshold-based semantic-segmented imaging data.** This analysis was conducted using the original, unprocessed output images with a custom-developed ImageJ[59] macro (access link available under Code Availability), with the following workflow summarized in Supplementary Fig. 23: First, the user was prompted to select a region of interest. We selected the regions corresponding to those used for all other data analyses in this study, excluding technical and imaging artifacts (Supplementary Fig. 4A). Next, ACTA2, CD144, and CD45 channel images were extracted. The CD45 channel was processed using ImageJ's Subtract Background function with a "Rolling ball" radius of 50 px. Each channel image (ACTA2, CD144, or CD45) was then subjected to a 2D Gaussian blur (radius: 4 px for ACTA2 and CD144; 2 px for CD45), followed by a semantic segmentation using ImageJ's Triangle threshold algorithm. Subsequently, masks derived from ACTA2 and CD144 channels were combined using the logical AND function, and a size filter was applied to remove all objects smaller than 2000 px. The resulting mask was presented to the user for correction if necessary (e.g. adding missing small arteries, or removing artifacts or airways detected incorrectly due to being too close to an artery). After user validation, this mask was expanded within the ACTA2 mask: to do this, a new image was generated, into which all objects from the ACTA2 mask were written if they had at least an overlap of 1 px with the mask validated by the user. Thereby, a mask was generated that did not only contain the centers of the arteries but extended to the ACTA2 region around the arteries. The user validated this expanded mask again, correcting it if needed. The validated mask was further expanded using a maximum filter with an 80 px radius, creating a mask that represented all areas within a 40 μm radius around arteries. Finally, this mask used to differentiate the immune cell mask (from the CD45 channel) into artery-close and artery-distant regions. The artery-close and the -distant masks were used to determine the mean and median intensity in artery-close and -distant immune cell regions in all image channels of the raw 30-plex image.

### Reporting summary

Further information on research design is available in the Nature Portfolio Reporting Summary linked to this article.

## Data availability

Processed fluorescent imaging datasets generated in this study in.qptiff file format, together with the most relevant set of metadata for each individual week, have been deposited to Zenodo: https://doi.org/10.5281/zenodo.11652584. Raw-image datasets (807 GB) are available from the corresponding authors on request because of data size limitations. Segmentation output files in.csv file format used for all the presented downstream analyses have been deposited to Zenodo: https://doi.org/10.5281/zenodo.11623168. Cell type annotations resulting from downstream analyses and marker intensity plots, both overlaid on DAPI channel images for each week, can be accessed under the "Spatial Proteomics" tab at the following Human Developmental Cell Atlas (HDCA) interactive portal: https://hdca-sweden.scilifelab.se/tissues-overview/lung/. Source data are provided with this paper.

## Code availability

Segmentation of fluorescence images was performed using PIPEX: https://github.com/CellProfiling/pipex, archived in Zenodo (https://doi.org/10.5281/zenodo.11642375). The code for all programmatic analysis performed in this study is available in the following GitHub repository: https://github.com/CellProfiling/HDCA-FetalLung-SpatialProteomics/, archived in Zenodo (https://doi.org/10.5281/zefishernodo.11650173).

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

## Acknowledgements

We acknowledge the service of the Karolinska Institutet Developmental Tissue Bank for providing the prenatal human tissues. The Spatial Proteomics Facility, funded by SciLifeLab and the National Microscopy Infrastructure NMI (VR-RFI 2016-00968) is acknowledged for technical support and providing access to instruments. We also acknowledge Christophe Avenel for the technical support regarding TissUUmaps, which was made possible in part by BioImage Informatics Facility, a unit of the National Bioinformatics Infrastructure Sweden NBIS, funded by the SciLifeLab, National Microscopy Infrastructure NMI (VR-RFI 2019-00217), and Chan Zuckerberg Initiative DAF (DAF2021-225261). Parts of the computations and data storage was enabled by resources provided by the Swedish National Infrastructure for Computing (SNIC) at UPPMAX partially funded by the Swedish Research Council (Vetenskapsrådet) (2018-05973). This work was supported mainly by grants to EL from the Knut and Alice Wallenberg Foundation (KAW 2018.0172) and the Erling Persson Foundation for the Human Developmental Cell Atlas (HDCA) project. BA acknowledges an Establishment Grant from the Swedish Research Council (Vetenskapsrådet) (2022-04732). JNH acknowledges a post-doctoral fellowship from the Wenner-Gren Foundations and an EMBO Postdoctoral Fellowship (ALTF 556-2022). This publication is part of the Human Cell Atlas—www.humancellatlas.org/publications/.

## Author contributions

S.S., B.A., A.M.C., E.L. developed the methodology for the study; E.S., X.L., Z.A., P.C. coordinated and/or performed the isolation and processing of human embryonic specimen; S.S. carried out the immunofluorescent experimental work; F.B.N., M.M., J.N.H. developed tools for analysis of imaging data; S.S., B.A., J.N.H. analyzed the data; S.M.S., J.N.H. provided support for data analysis; S.S., B.A., A.S., J.N.H., C.S., E.L. interpreted the data; S.S., B.A., J.N.H. created the figures; B.A., S.S., A.S. wrote the original manuscript; A.S., J.N.H., E.L. revised the manuscript; B.A., E.L. supervised and administered the project; E.L., B.A., E.S., J.L., M.N., C.S., S.L. acquired funding. All authors read and commented on the manuscript.

## Funding

## Competing interests

J.L. is a scientific consultant for 10X Genomics. SMS is a co-founder of Spatialist AB. The remaining authors declare no competing interests.
