## [Peer Review file · Nature Communications]

High-parametric protein maps reveal the spatial organization in early-developing human lung

Corresponding Author: Dr Burcu Ayoglu

Version 0:

Reviewer comments:

Reviewer #1

(Remarks to the Author)

Review of Sariyar et al Nat Comms 2024

High dimensional spatial analysis has provided many biological domains with cellular contact maps that help understand the physical organization of cells within tissues. As part of the Human Developmental Cell Atlas, Sariyar and colleagues seek to provide a unique and rare resource to the developmental biology community of multiplexed imaging with 30 markers on lungs from 5 first trimester human embryos. The authors take advantage of the rich information provided by 30 markers and analyze the resulting data set with respect to cell type composition, colocalization, and proliferation status. Interesting compositional patterns are pointed out including the ordered emergence of different immune populations and macrophage and T/ILC cell co-localization around blood vessels. Relative proliferation rates across time points also provide an insightful guide to temporal developmental patterns.

Resources such as this are important guides for follow up experiments and validation. To maximize its use, and the confidence of users, additional details should be provided and analysis done to support the claims made within the paper. Direct tests of specific claims within the data are sometimes not addressed, and the reader is sometimes left to interpret complex relationships from network diagrams and large heatmaps which can be hard to identify. Moreover, certain analyses appear to contain somewhat overlapping information and additional robustness analysis would add confidence to results regarding co-localization and spatial domains. Especially given the lower sample size, additional confidence in the resulting data and conclusions through analytical means and more targeted visualizations would greatly improve the manuscript.

Major points

1. Clustering.

- a. It is not clear why the data were first merged and then clustered and then re-clustered again for each time point. Please comment on the consistency of these two clusterings (comparing figure S4 to figure S5). Does batch effect correction affect the clustering results?
- b. The number of cells may not always be directly proportional to the area at different stages of development and the authors could report or comment on cells per unit area when comparing across tissue sections.

2. Domain annotation.

- a. Please add more support for the annotated domains provided in figure S9 and figure 4. The domains found seem to be mostly dominated by single cell types and their stability is unclear, and therefore it is unclear how much additional information they add beyond just the cell type labels. The results of UTAG have also been found to be less consistent in a recent comparison to other methods (Varonne et al Nature Genetics 2024). The robustness of called domains to input parameters could be assessed or compared to domain calls from areas of the tissue that were not used in this analysis.
- b. It is unclear how the domain adjacency results are calculated for figure 4B. All that is referenced is the 'Draw network' function but it is unclear where this comes from or what it is using.

3. Co-localization analysis.

- a. Please illustrate specific comparisons when these are referenced in the text (detailed in visualization minor point).
- b. Colocalization Z scores may be sensitive to the number of cells of a particular type and the number of neighbors. Please assess how the Z scores relate to the number of total interactions, the number of cells, and the number of neighbors used and if this influences the results. Additional metrics and significance tests (as in Figure S10C) to show colocalization between cell types would strengthen these claims. The analyses would also benefit from using a fixed radius threshold (provided in squidpy) so that physically distant nearest neighbors are not counted.
- c. Comparisons across timepoints are difficult to discern and when referenced in the text it would help to have a visual reflecting the specific comparison.
- i. One example is the proximity of mesenchymal Ki67+ cells to different cell types in figure 3. This statement is not clear from the heatmaps and could also be driven by adjacency between proliferating mesenchymal cells and mesenchymal cells of different subtypes, which may be related.
- ii. Also in figure 5D it is difficult to grasp claims that non-proliferating cell types having more homotypic patterns. Summary statistics across time points and grouped by proliferating and non-proliferating would help make this point.
- d. Are the claims from the domain co-localization supported at the cell type level?

4. Proliferation analysis.

- a. Please elaborate how cutoffs were defined for proliferating cells. Is the distribution bimodal? Are results consistent at different cutoffs?

5. Immune analysis.

- a. How is figure 7B different from figure S12C?
- b. How were random immune cells chosen across the tissue?
- i. Is a t-test of marker expression patterns consistent across random samples?
- c. Please report on significance of marker comparisons in T-tests.
- d. Are the same results w.r.t. marker expression around vessels observed when analyzed in a binary fashion by thresholding on marker positivity? This could help reduce the impact of segmentation errors on this result.
- e. Are there similar MHC-II high macrophage populations from the similar scRNA-seq study? And do these express endothelial remodeling factors such as VEGFA?

6. Methods details.

- a. It is unclear how normalization, background subtraction, and otsu thresholding were done in the first preprocessing for DAPI and EPCAM
- b. If just using EPCAM as a surface marker in segmentation, does this affect the quality of segmentation for immune and mesenchymal cells?
- c. No details provided for how Perseus was used to exclude non-lung areas or artifacts
- d. Comments on how the excluded regions would affect results would be helpful. Are there any cell populations that would be missing from certain weeks because of cropping?
- e. Example code would help facilitate interpretation of analyses and processing steps.

Minor points

1. Visualization.

- a. It is often difficult to interpret relative color intensities on the heatmaps or network diagrams as evidence for various claims throughout the manuscript. Where possible providing figures directly comparing co-localization Z-scores or cell distances would clarify the results. This would relate to claims such as the consistent colocalization of epithelium and ASM cells from week 8.5 on (Figure 3B), where from the heatmaps it only appears to be colocalized in 8.5 and 12. In addition it appears that the color scale for week 8.5 in figure 3B has a 0 value which is in the red area which could be misleading, although this could be a cropping issue.
- b. The color palette for figure 6E makes it difficult to distinguish different cell colocalizations.
- c. There are no error bars in figure S10C

2. Figure order

- a. Some of the supplementary figures are out of order with their position in the text.

Reviewer #2

(Remarks to the Author)

This well-written manuscript investigated the spatial organisation of human foetal lung cell types between 6-13 post conception weeks at protein level through high-parametric microscopy. An oligo-barcoded antibody panel (30 antibodies) representative of the main foetal lung cell types during the first trimester of gestation was designed and 1 million cells were analysed at single-cell resolution. First, different cell types, their main expressed markers and their abundance across developmental stages was determined. Next, the authors studied the adjacency patterns among the identified cell types, which highlighted that while cells of the same type predominantly shared adjacency, this pattern was dynamic throughout

developmental time points. Furthermore, they examined cellular interactions on a higher order by looking into multi-cellular units, suggesting an early emergence of lung tissue organisation, and highlighting the importance of spatial organisation of cells and their interactions in orchestrating lung development. They also explored the composition of the different proliferative cell types across developmental stages, and their adjacency patterns, indicating a temporal sequence in lung development, as well as a spatial gradient of proliferation along the proximal-distal axis of the developing epithelium. Lastly, they delved into the immune landscape, reporting on the abundance of various immune cells across developmental timepoints, their spatial proximity patterns with other immune cells, and their spatial arrangement patterns within the tissue.

This would be an extremely valuable resource to have, but based on the following comments, I do not think it is ready for publication in its current form.

MAJOR POINTS

- 1) The number of biological replicates is only 1 per 5 timepoints. One time-point in the embryonic and 4 in the pseudoglandular stages with no mention of technical replicates. This is clearly insufficient.
- 2) I cannot see any statistical analyses done.
- 3) The paper is very descriptive with few if any novel findings.
- 4) I acknowledge the effort used to validate the 30 markers, but nevertheless it is very difficult to reliably identify granular subtypes using only 30 markers across epithelial, mesenchymal and immune compartments.

MINOR POINTS

- 1) Why are some cell types missing, e.g. a) mesothelial cells only present at 6pcw and then not, b) neurons only detected at 11pcw?
- 2) 'Dendritic cells, NK cells and B cells emerged from 11 weeks onwards (Figure 6B) emphasizing the distinct timelines of immune cell formation during development.' - In Barnes et al., Sci Immuno 2023 paper, all of these cell types were present from 8 weeks according to scRNAseq; plus immunohistochemistry shows DCs at 8 pcw.
- 3) Overlapping numbers on UMAP in Fig S5A.
- 4) Figure 6, no CD8 T-cells identified.
- 5) Why was 50microns chosen as the radius to use for looking at immune cells around arterial areas. This seems quite a large distance.
- 6) Fig 4C – it is hard to differentiate between some of the lines on this plot, especially where colours are so similar e.g. week 11 plot.
- 7) Figure 5A and B – IHC images in 5A show only a fraction of the results presented in 5B. It is important to include all IHC images that support results for the composition of proliferative cells shown (e.g. Mesenchymal, SOX2^{high} epithelium, SOX9^{high} epithelium, lymphatic endothelium, ASM, VSM), at least somewhere in the supplementary information.
- 8) General comment. The colours are too similar to be able to differentiate between the cell types in these plots, such as Figure 6C, so the colours need changing, to allow this. If that doesn't help, then the plots should be removed, as you can't discern any useful information from them. This also applies to Supplementary Figure 7 (too many similar shades of blue), Figure g (B,D,E), Figure 5B (cannot distinguish between Mesenchymal and ASM cells).

Reviewer #3

(Remarks to the Author)

The manuscript by Sariyar et al. aims to unravel the cellular heterogeneity and cellular protein landscape in the developing respiratory system by exploring the cell phenotype in the lungs, trachea, and vasculature. As part of a Human Developmental 7 Cell Atlas (HDCA) initiative, the authors describe a spatiotemporal organization of lung cells at different time points of development during the first trimester by applying a high-parametric tissue imaging approach using a 30-plex antibody panel. Using this novel technology, the authors define cell phenotypes of individual lung cells across five developmental timepoints in the first trimester, with a comprehensive spatially resolved cell type composition, rate of cell proliferation and cellular spatial location. These data are very insightful and provide an excellent resource for studies investigating development of the respiratory immune system in prenatal and early life. Furthermore, this manuscript could serve as an essential platform for interrogating human respiratory health and disease. Overall, the data provided is very impressively carried out with some sections needing further clarity. Some of the points highlighted are below:

Major points:

1. The authors use DAPI as a nuclear marker and EPCAM as a membrane marker however EPCAM is not expressed on non-epithelial cells? So how confident can the authors be with segmenting all the lung cells accurately?
2. The authors describe Sox9^{high} and Sox2^{high} epithelial cells but do not show any marker trajectories for ciliated, secretory and neuroendocrine cells. This would be important to understand how physiological epithelial differentiation and turnover occurs during development.
3. The authors analyse each condition separately and extract data out to perform statistical analysis between conditions. This also explains why there are so many graphs generated as they would need one set of graphs for every condition. In this case it is quite difficult to make strong conclusions purely because you can't statistically compare one network graph to another, which at the moment as I understand are selected based on visual indication. For example, in figure 3 the authors have all of their correlation matrices, where you can see some small differences perhaps, but these graphs do not show a statistical significance between the weeks so I don't think you could say there are

significant difference between the matrices. One would have to pull out the individual Z scores, plot them and do statistical testing of those to get anything definitive.

4. Also it is unclear how many donor tissues were obtained per developmental time point and whether the clinical status of the donating mother was assessed. These would strongly impact lung developmental cell states.

Version 1:

Reviewer comments:

Reviewer #1

(Remarks to the Author)

I commend the authors for doing many analyses to support and strengthen the manuscript, and most of my comments have been addressed. I particularly find the line plots for colocalization, the comparison to scRNA-seq, and a more clear comparison of proliferation differences to be great additions. I also believe their discussion of limitations and nuanced technical aspects to be greatly informative to the reader when assessing the data.

Minor comments

- The claim that neuronal cells had increased adjacency with ASM cells from week 8.5 on doesn't appear to make sense because I don't see neurons identified in week 6.
 - It is difficult to say the cell density is stable without replicates. It can be said it is generally consistent although increases slightly in later weeks
 - The claim that the domains cluster based on type rather than time point is very difficult to assess given the lack of replicates and it is not clear that that is the case based on the correlation heatmap
- The mesenchymal-immune rich domain clusters segregate into two groups based on time point proximity

Reviewer #2

(Remarks to the Author)

Thank you for submitting a detailed response to my comments.

MAJOR POINTS

- I understand the difficulty in obtaining intact human fetal lungs, but using only 1 biological replicate per time points with 5 points in total and only 1 technical replicate per time point is simply not enough to give confidence in the results. I am also not sure how statistics was done on only 1 technical replicate at each time point. I appreciate all the hard work which has been done so far, but to me more biological replicates are needed before considering publication. At the very least more technical replicates are needed. Would a transfer to a methods journal be more appropriate?

MINOR POINTS

- In the Methods section, no karyotype analysis was done. This would be very important in view of only 1 sample being used per time point.
- It seems that classifying DCs was mainly based on HLA-DR expression, but this is not an exclusive marker of DCs.
- Page 5: "Fine" annotation is not appropriate to use as it only gives broad cell types. Classification is only based on 1 or 2 markers.
- The authors state that NK cells and B cells are the least abundant cells during early development. However, the authors' own single-cell data (Nature Cell Biology, 2023) show that there are plenty of NK cells. So why are they now not seeing NK cells in their data set – perhaps this needs looking at in more detail, using more markers and in more tissue sections?

Authors' Response to Reviewer Comments

We are immensely grateful to all three reviewers for their insightful and constructive feedback. We appreciate the time and effort they invested in reviewing our work. Their expert guidance provided us to strengthen the manuscript. We have diligently addressed and thoroughly discussed each question/comment. As indicated here and in the revised manuscript file (with tracked changes), we implemented several changes in the main text file. In response to reviewers' feedback, all main figures except Figure 1 were also changed significantly, since e.g. the analyzed region for the 8.5-week-old was changed and several spatial analyses were repeated with changed parameters/approaches. Also, a new set of 20 figures were prepared as supplementary figures.

Reviewer #1 (Remarks to the Author):

High dimensional spatial analysis has provided many biological domains with cellular contact maps that help understand the physical organization of cells within tissues. As part of the Human Developmental Cell Atlas, Sariyar and colleagues seek to provide a unique and rare resource to the developmental biology community of multiplexed imaging with 30 markers on lungs from 5 first trimester human embryos. The authors take advantage of the rich information provided by 30 markers and analyze the resulting data set with respect to cell type composition, colocalization, and proliferation status. Interesting compositional patterns are pointed out including the ordered emergence of different immune populations and macrophage and T/ILC cell co-localization around blood vessels. Relative proliferation rates across time points also provide an insightful guide to temporal developmental patterns.

Resources such as this are important guides for follow up experiments and validation. To maximize its use, and the confidence of users, additional details should be provided and analysis done to support the claims made within the paper. Direct tests of specific claims within the data are sometimes not addressed, and the reader is sometimes left to interpret complex relationships from network diagrams and large heatmaps which can be hard to identify. Moreover, certain analyses appear to contain somewhat overlapping information and additional robustness analysis would add confidence to results regarding co-localization and spatial domains. Especially given the lower sample size, additional confidence in the resulting data and conclusions through analytical means and more targeted visualizations would greatly improve the manuscript.

We are grateful to Reviewer 1 for their comprehensive review of our initial submission. We truly appreciate their constructive comments, and their thoughtful and well-structured feedback, which was essential in guiding our revisions. The suggestions for enhancing data confidence and clarity through more targeted visualizations and analytical rigor were invaluable to improve the overall quality of our work. We addressed these now in the revised version of our work, and please see our point-by-point responses and the changes made to the manuscript below.

Major points

1. Clustering.

a. It is not clear why the data were first merged and then clustered and then re-clustered again for each time point. Please comment on the consistency of these two clusterings (comparing figure S4 to figure S5). Does batch effect correction affect the clustering results?

We acknowledge that our initial submission lacked a clear explanation for our use of the merged dataset encompassing all ages, as well as for the dataset from individual timepoints. To address that, we provided a flowchart in the revision as **Supplementary Figure 5**, also shown to the left. As summarized in this flowchart, we first investigated each timepoint for possible artifact clusters spatially corresponding to imperfections in the images. After removal of cells corresponding to these clusters, we merged the data from all timepoints using a batch correction method called ComBat¹ to gain a preliminary insight into the main cell types identifiable in this developmental tissue type with our antibody panel. However, aware of the limitations inherent to batch correction methods - specifically, their dependence

¹ Johnson W., *et al.* Adjusting batch effects in microarray data using empirical bayes methods. *Biostatistics*. 8:118–127 (2007).

on consistent cell type and state compositions consistent across “batches”²- we proceeded with finer annotations of clusters identified as the result of clustering individual timepoints. This was also based on a comparison of cell type annotations for each cell when clustered using either the merged (batch-corrected) or individual timepoint data (shown below and as **Supplementary Figure 11**). This comparison revealed that while a majority (78%) of the cell type assignments were consistent, discrepancies existed in the assignment of certain cell types such as SOX2^{high} epithelial cells, immune cells and airway smooth muscle cells. Further examination of these discrepancies revealed that the cell type assignments based on individual timepoint data were more accurate than those based on the batch-merged data, indicating that batch correction may introduce distortions in cell type assignments. Consequently, all our in-depth downstream analyses have been conducted using datasets from individual timepoints rather than the merged data.

We now included this explanation in the revision:

Page 21, line 14:

“A comparison of cell type annotations for each cell when clustered using either the merged or individual timepoint data revealed that while a majority (78%) of the cell type assignments were consistent, discrepancies existed in the assignment of certain cell types, such as SOX2^{high} epithelial cells, immune cells and airway smooth muscle cells (Supplementary Figure 11). Further examination

of these discrepancies revealed that the cell type assignments based on individual timepoint data were more accurate than those based on the merged data. Consequently, all in-depth downstream analyses were conducted using datasets from individual timepoints rather than the merged data.”

² Luecken M.D and Theis F.J. Current best practices in single-cell RNA-seq analysis: a tutorial. Mol Syst Biol. 15(6):e8746. doi: 10.15252/msb.20188746 (2019).

b. The number of cells may not always be directly proportional to the area at different stages of development and the authors could report or comment on cells per unit area when comparing across tissue sections.

Initially, we were curious about this detail because we hypothesized that there might be a change in cell density during this developmental window. Using the calculated area of the polygons depicted in **Supplementary Figure 4A**, we calculated the cell density. Interestingly, this revealed a robust temporal trend despite slight fluctuations, with an average of 2,564 cells per mm² across the different developmental weeks. In our initial submission, we chose not to include this detail, but we added it now in table format as **Supplementary Figure 4C** to the revised manuscript along with a brief comment on the finding.

Page 4, line 10:

“Interestingly, we observed a stable cellular density across the different developmental weeks (Supplementary Figure 4C).”

2. Domain annotation.

a. Please add more support for the annotated domains provided in figure S9 and figure 4. The domains found seem to be mostly dominated by single cell types and their stability is unclear, and therefore it is unclear how much additional information they add beyond just the cell type labels. The results of UTAG have also been found to be less consistent in a recent comparison to other methods (Varonne et al Nature Genetics 2024). The robustness of called domains to input parameters could be assessed or compared to domain calls from areas of the tissue that were not used in this analysis.

We are grateful for this crucial feedback. Our initial plan of action to address this comment was to re-assess the robustness of domains identified with the UTAG algorithm³ by increasing its radius parameter (from 12 to 30) and by repeating the domain analysis across additional selected areas of the tissue, as the Reviewer described. We have indeed performed these analyses. However, UTAG presented still one major

³ Kim, J. *et al.* Unsupervised discovery of tissue architecture in multiplexed imaging. Nature Methods 19, 1653-1661, doi:10.1038/s41592-022-01657-2 (2022)

disadvantage for our purpose: As the referee correctly pinpoints, UTAG allowed analysis of only portions of the tissue area in our large sections of 11 to 13 weeks. It was not scalable to our type of large tissue dataset consisting of up to 350k cells in each week. We therefore changed our plan of action and investigated alternative algorithms for unsupervised spatial domain analysis. We were not able to implement the more recent algorithm referenced by the reviewer, namely CellCharter⁴, however the collective expertise among the co-authors guided us towards another recent algorithm called BANKSY (Building Aggregates with a Neighborhood Kernel and Spatial Yardstick)⁵, whose performance some of the co-authors of this manuscript recently benchmarked for Xenium *in situ* sequencing (ISS) data⁶. In addition to its reported performance metrics compared to other available algorithms for segmenting tissue domains, including also its application on a highly similar spatial proteomics dataset as ours, it is also the most scalable and fastest among several other existing methods, and in our hands, BANKSY allowed domain identification analysis in whole tissue sections. In the revised version of our manuscript, we therefore opted to base the unsupervised identification of spatial domains on use of the BANKSY algorithm. The new results of domain analysis are shown in **Figure 4**. These are reported and discussed as follows:

Page 7, line 25:

*“Mesenchymal cells were present in almost all identified spatial domains across all developmental weeks, except the epithelium- and chondroblast-rich domains, where the highest mesenchymal cell content was below 4%. SOX2^{high} epithelium-, SOX9^{high} epithelium-, and chondroblast-rich domains were more homogeneous compared to other spatial domains, which were dominated by the corresponding individual cell types, constituting 59-89% of these domains (**Figure 4, Supplementary Figure 15**). This indicates the topographic segregation of these domains during the early formation of the organ. The most noticeable temporal change in the composition of the domains was that in 6-week-old lung, SOX2^{high} epithelial cells and SOX9^{high} epithelial cells constituted one single spatial domain, whereas from week 8.5 onwards, these cell types dominated their individual spatial domains.*

Hierarchical clustering of the correlation across all individual spatial domains across all developmental weeks was driven by the type of domains, rather than the

⁴ Varrone, M. *et al.* CellCharter reveals spatial cell niches associated with tissue remodeling and cell plasticity. *Nat Genet* 56, 74–84 (2024)

⁵ Singhal, V., *et al.* BANKSY unifies cell typing and tissue domain segmentation for scalable spatial omics data analysis. *Nat Genet* 56, 431-441 (2024)

⁶ Salas, S.M. *et al.* Optimizing Xenium In Situ data utility by quality assessment and best practice analysis workflows. *bioRxiv* 2023.02.13.528102 (2023)

developmental age. Airway smooth muscle cell-rich domains, chondroblast-rich, and SOX2^{high} and SOX9^{high} epithelium-rich domains from all weeks clustered separately among each other, whereas the relatively more mixed mesenchyme-, endothelium-, and immune- rich domains were forming another cluster while still being driven by the domain types rather than the developmental age (Supplementary Figure 16). Thus, the unsupervised domain analysis confirmed the previously described cell type adjacency patterns among mesenchymal, endothelial and immune cells. In summary, our data suggest the presence of recurring organizational domains emerging in the developing human lung as early as in week 6 to form the blueprint for a mature lung tissue organization. ”

We also updated the Methods section accordingly:

Page 22, line 17:

“For unsupervised analysis of the tissue architecture by identification of tissue domains, BANKSY algorithm³⁶ was used based on its comparative performance against similar algorithms³⁷. Tissue domains were independently identified in each developmental week by using the mixing parameter $\lambda=0.8$. After this, domains were annotated based on their cellular composition (Figure 4). Cell type composition within each domain was calculated as percentages (Supplementary Figure 15) and similarity between domains across different weeks was computed by calculating the Pearson’s correlation coefficient (Supplementary Figure 16).”

b. It is unclear how the domain adjacency results are calculated for figure 4B. All that is referenced is the ‘Draw network’ function but it is unclear where this comes from or what it is using.

In our initial submission, the domains were identified using the UTAG algorithm/package³, which also offered a function to calculate and visualize a domain adjacency matrix. Citing the UTAG publication itself: “The results [of this domain adjacency analysis] are presented as a network⁷ (v.2.6.2) graph in a spring force layout, which visually demonstrates how each domain colocalizes with others. This was done on the logarithm of the counts of edge connections to ensure that the counts are on a comparable scale.”

However, as reasoned and described above, we now changed the tool for the unsupervised identification of spatial domains and opted also to remove the domain adjacency analysis where the previously identified spatial domains using the UTAG algorithm were used. We believe such an analysis with domains generally containing

⁷ Aric A. *et al.* Exploring network structure, dynamics, and function using NetworkX, Proceedings of the 7th Python in Science Conference (SciPy2008) pp. 11–15 (2008)

more mixed cell types is not relevant anymore and we preferred to focus the discussion more on the cell type content of identified domains. Consequently, this analysis and related sections have been removed in the revised version.

3. Co-localization analysis.

a. Please illustrate specific comparisons when these are referenced in the text (detailed in visualization minor point).

This was a very useful feedback. In order to address this in the revised version, we first scaled and combined the neighborhood enrichment Z-scores obtained in each week in order to provide a more comparable overview of the cell type adjacency patterns across the different weeks (**Figure 3B**). In addition, for the main cell types commonly identified and annotated across all weeks, we provided line plots (shown left), providing a better reference visual summarizing the temporal changes in their scaled enrichment Z-scores for neighboring their own or the rest of the other cell types (**Supplementary Figure 14**).

We then also merged the different timepoints for each main cell type to compare the degree of homotypic neighboring patterns for proliferating and non-proliferating cells, as shown below (**Supplementary Figure 18**). We believe

these changes now allow for a more straightforward comparison of the spatiotemporal trends observed across the different weeks.

b. Colocalization Z scores may be sensitive to the number of cells of a particular type and the number of neighbors. Please assess how the Z scores relate to the number of total interactions, the number of cells, and the number of neighbors used and if this influences the results. Additional metrics and significance tests (as in Figure S10C) to show colocalization between cell types would strengthen these claims. The analyses would also benefit from using a fixed radius threshold (provided in squidpy) so that physically distant nearest neighbors are not counted.

In order to study cell type adjacency patterns, instead of a selected number of neighbors, we now incorporated a fixed radius threshold, as the reviewer suggested. By doing this, we believe we also address the concern that the neighborhood enrichment patterns might be biased for certain cell types. By using a fixed radius threshold, each cell's neighborhood is determined by the same spatial criteria, irrespective of the local cell density or the total number of cells in a cluster. Since all neighborhoods are defined by the same spatial criteria, the Z-score values of the enrichment statistics should not be influenced by the varying number of cells in different clusters (in contrast to enrichment counts themselves). Also, enrichment Z-scores inherently account for the variability in interaction counts due to random chance, providing a robust measure of enrichment that is not biased by the number of cells in each cluster.

We updated the related Methods section:

Page 21, line 28:

“Spatial localization patterns of annotated cell type clusters were analyzed using spatial neighbors function (sq.gr.spatial_neighbors(adata, radius=50, coord_type="generic")) in Squidpy³⁵, which creates a graph from spatial coordinates based on a selected fixed radius, which was 50 for this analysis.”

c. Comparisons across timepoints are difficult to discern and when referenced in the text it would help to have a visual reflecting the specific comparison.

To address this concern in the revised version, we scaled the neighborhood enrichment Z-scores obtained in each week and combined them in a dataframe in order to provide a more comparable overview of the cell-cell adjacency patterns across the different weeks (**Figure 3B**). In addition, for the main cell types common across all weeks, we provided visuals summarizing the temporal changes in their scaled enrichment Z-scores for neighboring their own or the rest of the other cell types (**Supplementary Figure 14**). We believe these changes allow now for a more straightforward comparison of the cellular adjacency patterns for each main cell type with itself or the rest of the other cell types across timepoints.

i. One example is the proximity of mesenchymal Ki67+ cells to different cell types in figure 3. This statement is not clear from the heatmaps and could also be driven by adjacency between proliferating mesenchymal cells and mesenchymal cells of different subtypes, which may be related.

We acknowledge that this statement was initially not supported directly in any of the visuals. As discussed for the previous point, we believe basing our analysis and visualizations on scaled enrichment Z-scores in this revised version generally addresses this concern. Then, the new **Supplementary Figure 14**, as also mentioned for the previous comment, now provides a visual aid to support the specific statement regarding Ki67+ mesenchymal cells: Airway smooth muscle cells, endothelial cells, immune cells, SOX2^{high} and SOX9^{high} epithelial cells all demonstrate clear homotypic cellular adjacency patterns dominating over their adjacency patterns with the other cell types, whereas in particular Ki67+ mesenchymal cells, but also Vim+ mesenchymal cells, demonstrate a slightly more mixed cellular adjacency pattern, where they share adjacency e.g. with endothelial cells and immune cells. However, we agree with the reviewer that we should rephrase this statement. In the revised manuscript, this has been changed:

Page 6, line 26:

*“Compared to other cell types, mesenchymal cells, including proliferating mesenchymal cells, neighbored more frequently with cells of other types, indicating their dispersed distribution throughout the tissue space during the early stages of human lung development (**Supplementary Figure 14**).”*

ii. Also in figure 5D it is difficult to grasp claims that non-proliferating cell types having more homotypic patterns. Summary statistics across time points and grouped by proliferating and non-proliferating would help make this point.

We appreciate this suggestion. In the revised manuscript, instead of the heatmaps in **Figure 5D**, we now provide a more thorough analysis and summary statistics, summarized in **Supplementary Figure 18**, to support the general observation that non-proliferating cell types have more homotypic adjacency patterns than their proliferating counterparts. Referring to **Supplementary Figure 18A** (below), when merging all the timepoints, for airway smooth muscle cells, SOX2^{high} epithelial cells, endothelial cells and immune cells we observe more homotypic adjacency patterns in non-proliferating cells than in their proliferating counterparts (Student's t-test p-values<0.05). A similar trend (with no statistically significant difference) was observed, for SOX9^{high} epithelial cells, and interestingly, an opposite trend (with no statistically significant difference) was observed for mesenchymal cells, which might suggest the presence of dense niches of mesenchymal proliferation in the developing human lung.

We now summarized these analyses in the revised manuscript:

Page 9, line 12:

“A cumulative analysis across all weeks indicated that non-proliferating airway smooth muscle cells, immune cells, endothelial cells, SOX2^{high} epithelial cells, and SOX9^{high} epithelial cells, ordered by statistical significance, exhibited significantly more homotypic adjacency patterns compared to their proliferating counterparts (Supplementary Figure 18A). Conversely, mesenchymal cells showed the opposite trend, with proliferating mesenchymal cells displaying a higher degree of

homotypic cellular adjacency pattern compared to their non-proliferating counterparts, which might imply the presence of centralized niches of mesenchymal proliferation during early human lung development.”

d. Are the claims from the domain co-localization supported at the cell type level?

As also noted by the reviewer in a previous comment regarding domain analysis, the domains identified using UTAG algorithm within limited, selected regions of the tissues were dominated by single cell types, and therefore the main focus when describing these results were concentrated more on the proximity relations between these domains. Revisiting this analysis now in whole tissue sections with another algorithm, BANKSY, provided new sets of domains with generally more mixed cell type compositions reflecting more accurately the recurring organizational motifs and thus the organ-scale task delegation in the developing human lung. As mentioned previously, the analysis of proximity relations among the domains was not relevant anymore and therefore has been removed in the revised version.

4. Proliferation analysis.

a. Please elaborate how cutoffs were defined for proliferating cells. Is the distribution bimodal? Are results consistent at different cutoffs?

We are grateful for this comment as it allowed us to provide a more accurate description of our approach to define intensity thresholds for the proliferation marker Ki67. To determine the intensity thresholds, we inspected the images and the normalized Ki67 intensity histograms in each week. Inspecting the images and looking at Ki67 signals in different cell types, we observed that we were not detecting a Ki67 signal for the majority of the cells, and that intensities for the Ki67-positive cells were spread across a larger intensity range. For each week, we visually inspected the images (as provided now in **Supplementary Figure 17**) and determined intensity thresholds that distinguished the Ki67-negative and Ki67-positive cells. We next aimed to validate these thresholds by

computing the intensity histograms, median values, and 75, 85, and 95th percentiles for each week:

Comparing our thresholds and the histograms, we observed that our manually selected intensity thresholds were close to the lowest intensity bin, which represents a strong peak collecting the clearly “negative” cells. Our thresholds corresponded to the 85th percentile for weeks 11 and 12; the 70th percentile for weeks 6 and 8.5, and the 86th percentile for week 13, as shown above.

Statistics for Week 6 sample:
 Peaks at positions: [-0.03262878 0.95274016]
 Number of peaks: 2
 The data appears to be bimodal (multimodal).

Statistics for Week 11 sample:
 Peaks at positions: [1.89043660e-04 5.98020752e-01 7.02404066e-01 7.78319203e-01 8.50016833e-01 1.00922775e+00]
 Number of peaks: 6
 The data appears to be multimodal.

Statistics for Week 8.5 sample:
 Peaks at positions: [7.63020901e-04 1.07432763e+00]
 Number of peaks: 2
 The data appears to be bimodal (multimodal).

Statistics for Week 12 sample:
 Peaks at positions: [0.00104045 0.3886072 0.42877727 0.51878797]
 Number of peaks: 4
 The data appears to be multimodal.

Statistics for Week 13 sample:
 Peaks at positions: [0.00135656 0.8537064]
 Number of peaks: 2
 The data appears to be bimodal (multimodal).

We also confirmed that there were at least two populations of cells present (bimodal distribution) by performing Kernel Density Estimation using Gaussian kernels to estimate the probability density function in a non-parametric way and automate bandwidth determination. As shown in the results above, our data appeared to be multimodal with a more graded than binary pattern, which aligns well with the findings from Miller *et al.* (2018) regarding Ki67⁸. In all weeks, there is a peak representing the "Ki67-negative" cells and one or multiple peaks for "Ki67-positive cells". This confirms our strategy to delineate positive and negative cells based on thresholds close to the peak for Ki-67 negative cells. These peaks were located to the left of the manually selected intensity

⁸ Miller I. *et al.* Ki67 is a graded rather than a binary marker of proliferation versus quiescence. Cell Rep. Jul 31;24(5):1105-1112.e5. doi: 10.1016/j.celrep.2018.06.110. (2018)

thresholds, confirming that we indeed split the cells into “negative” and “positive” cells by applying the selected thresholds.

In the revised version of our Methods section, we provided the following details to offer a more detailed explanation regarding this point:

Page 22, line 27:

“To determine proliferation patterns, we used the background-subtracted data frame for all weeks. For each individual week, we determined a Ki67 threshold value that separated the cell population of that week into Ki67-positive and -negative cells based on the image data. The thresholds corresponded to the 85th percentile for weeks 11 and 12, the 70th percentile for weeks 6 and 8.5, and the 86th percentile for week 13. Based on these intensity thresholds, Ki67+ cells in the Anndata object were annotated as ‘proliferating’.”

5. Immune analysis.

a. How is figure 7B different from figure S12C?

In **Figure 7B**, we compared the immediate local neighborhood composition of artery-close immune cells to that of other artery-distant immune cells selected randomly and in equal numbers. In (previously) **Supplementary Figure 12C** (now **Supplementary Figure 22D**), the comparison was similar but included cells of any type, not only immune type, which were again chosen randomly and matching the artery-close immune cells’ quantity. We reviewed the figure legends and think the differences between the two analyses are clearly explained, but we now elaborated on these two analyses in the “Comparative analyses of immune cells encircling vasculature” section of Methods:

Page 24, line 30:

“The immediate cellular neighborhood composition of the “artery-close” immune cells were compared both to randomly selected other, i.e. “artery-distant” immune cells, as well as to cells of any type, matching in both cases the artery-close immune cells’ quantity.”

b. How were random immune cells chosen across the tissue?

To select random, artery-distant immune cells (or also random cells of any type) in order to compare their local neighborhood composition with those of artery-close immune cells, we used Python's `sample()` function. This function randomly selected artery-distant

immune cells (or also random cells of any type) from the entire pool, excluding artery-close immune cells, in a quantity matching the number of artery-close immune cells. We now mentioned this in the “Comparative analyses of immune cells encircling vasculature” section of Methods:

Page 25, line 9:

*“The same analysis was performed for the equal number of artery-distant immune cells (**Figure 7B**), and also equal number of cells of any type (**Supplementary Figure 22D**), by randomly selecting them using `sample()` function.”*

i. Is a t-test of marker expression patterns consistent across random samples?

We acknowledge a potential misunderstanding concerning the marker expression analysis between artery-close cells and artery-distant immune cells. To clarify, the referred comparative marker expression analysis involved *all* cells annotated as artery-close immune cells and *all* cells annotated as other artery-distant immune cells, and not only a selection of the latter. A random set of artery-distant immune cells were chosen to match the number of artery-close immune cells only in the above discussed analysis for comparison of their local neighborhood composition. Given this clarification, the concern raised seems to be addressed. We have also revised the 'Comparative analyses of immune cells encircling vasculature' section in the Methods to state this more clearly.

Page 25, line 14:

“In addition to the comparison of their neighborhood composition, average marker expression patterns of all artery-close and all artery-distant immune cells were compared. For this differential expression analysis between artery-close and -distant immune cells, a two sided Wilcoxon rank-sum test was applied. [...]”

c. Please report on significance of marker comparisons in T-tests.

In the revised version of the manuscript, we provide the below shown Volcano plots in **Figure 7C**, where fold change (*i.e.* \log_2 of the ratio of mean marker expression in “Artery-close immune cells” to mean marker expression in “Artery-distant immune cells”) is plotted against $-\log_{10}$ of Benjamini-Hochberg corrected p-values obtained in Wilcoxon rank-sum test for all markers. We believe this provides a more accurate overview of the differential expression of markers between arterial and other immune cells. As stated for the previous comment, the related Methods section has also been revised.

d. Are the same results w.r.t. marker expression around vessels observed when analyzed in a binary fashion by thresholding on marker positivity? This could help reduce the impact of segmentation errors on this result.

We performed this new analysis suggested by the reviewer, where we did not use the nucleus-based instance segmentation, but instead used an intensity-threshold-based semantic segmentation for the vessel markers CD144 and ACTA2 and for the immune cell marker CD45 to measure marker intensities of artery-close immune cell regions.

Briefly, for the 11- to 13-week-old samples, we created masks for developing arteries using intensity-threshold-based semantic segmentation of the CD144 and ACTA2 channel, and we created masks for immune cell regions using intensity-threshold-based semantic segmentation of the CD45 channel. Finally, we used the artery mask to discern artery-close and -distant immune cell regions in the CD45 mask. We then extracted and compared the expression values for all markers of the panel in artery-close and -distant regions. More details are provided in **Supplementary Figure 23** (shown also above) and in the Methods section.

Page 25, line 25:

“Comparative analyses using intensity threshold-based semantic-segmented imaging data:

*This analysis was conducted using the original, unprocessed output images with a custom-developed ImageJ⁴¹ macro (access link available under Code Availability), with the following workflow summarized in **Supplementary Figure 23**: First, the user was prompted to select a region of interest. We selected the regions corresponding to those used for all other data analyses in this study, excluding technical and imaging artifacts (**Supplementary Figure 4A**). Next, ACTA2, CD144, and CD45 channel images were extracted. The CD45 channel was processed using ImageJ’s Subtract Background function with a “Rolling ball” radius of 50 px. Each channel image (ACTA2, CD144, or CD45) was then subjected to a 2D Gaussian blur (radius: 4 px for ACTA2 and CD144; 2 px for CD45), followed by a semantic segmentation using ImageJ’s Triangle threshold algorithm. Subsequently, masks derived from ACTA2 and CD144 channels were combined using the logical AND function, and a size filter was applied to remove all objects smaller than 2,000 px. The resulting mask was presented to the user for correction if necessary (e.g. adding missing small arteries, or removing artifacts or airways detected incorrectly due to being too close to an artery). After user validation, this mask was expanded within the ACTA2 mask: to do this, a new image was generated, into which all objects from the ACTA2 mask were written if they had at least an overlap of 1 px with the mask validated by the user. Thereby, a mask was generated that did not only contain the centers of the arteries but extended to the ACTA2 region around the arteries. The user validated this expanded mask again, correcting it if needed. The validated mask was further expanded using a maximum filter with an 80 px radius, creating a mask that represented all areas within a 40 μm radius around arteries. Finally, this mask used to differentiate the immune cell mask (from the CD45 channel) into artery-close and artery-distant regions. The artery-close and the -distant masks were used to*

determine the mean and median intensity in artery-close and -distant immune cell regions in all image channels of the raw 30-plex image.”

As summarized in the plot below, now **Figure 7D**, and a **Supplementary Table 4**, this analysis revealed a highly similar group of markers having higher or lower average marker expression pattern: We observed that CD90, HLA-DR and CD31 signals were increased, and CD68, CD163 and MRC1 signals were decreased in the artery-close immune-cell regions compared to the artery-distant immune-cell regions. We referred to these results in the revised manuscript:

Page 11, line 9:

“Differential marker analysis between the artery-close and artery-distant immune cells revealed that artery-close immune cells exhibited relatively higher levels of markers such as CD90 and HLA-DR, and relatively lower levels of CD163 and MRC1 (Figure 7C). A complementary image analysis approach using intensity threshold-segmentation (Supplementary Figure 23) confirmed these differential expression patterns (Figure 7D).”

e. Are there similar MHC-II high macrophage populations from the similar scRNA-seq study? And do these express endothelial remodeling factors such as VEGFA?

This is a great comment. We reviewed the scRNA-seq human developmental lung dataset published in the related previous work by Sountoulidis *et al*, 2023⁹ to identify similar macrophage subpopulations with potential roles in vascular growth and remodeling. In this dataset, which includes developmental weeks from 5 to 14, three clusters were initially annotated as macrophages, comprising a total of 905 cells. The heatmap to the left, now **Supplementary Figure 24**, summarizes the expression of 13 genes in these cells: VEGFA, VEGFB, CD45/PTPRC and eleven MHC Class II protein complex encoding genes for HLA-DR, HLA-DQA, HLA-DP, HLA-DM, and HLA-DOA. There is indeed a relatively small population of cells co-expressing both VEGFA and one or more of the HLA genes. In our protein- and image-level observations, these cells, arranged in a special pattern encircling vessels, are not abundant. Therefore, the small fraction of cells co-expressing these markers at the transcriptome level aligns with our protein- and image-level observations. We now referred to this in our revision:

Page 11, line 14:

“Concurrently, expression patterns for the corresponding genes, as well as for vascular remodeling factors such as VEGF-A within macrophage populations identified in single-cell transcriptomics data reported by Sountoulidis et al. 2023⁷ further supported the presence of distinct immune cell subpopulations potentially involved in vascular remodeling (Supplementary Figure 24)”

⁹ Sountoulidis, A. *et al.* A topographic atlas defines developmental origins of cell heterogeneity in the human embryonic lung. *Nature Cell Biology* 25, 351-365, doi:10.1038/s41556-022-01064-x (2023).

6. Methods details.

a. It is unclear how normalization, background subtraction, and otsu thresholding were done in the first preprocessing for DAPI and EPCAM.

In the revised version of our manuscript, we now provide further details about this point in the “Image processing and segmentation” sub-section under Methods.

Page 19, line 7:

“These single-channel images were then uploaded to our in-house developed software PIPEX (<https://github.com/CellProfiling/pipex.git>) to perform instance segmentation and various downstream single-cell data analysis. For this, DAPI and EPCAM marker images were preprocessed as follows to identify the most significant signal intensity thresholds: First, background subtraction was performed by subtracting the lower 0.01 percentile from the signal intensity values. This was followed by a min-max normalization, where the minimum value was transformed into a 0 and the maximum value was transformed into a 1, and every other value was transformed into a decimal between 0 and 1. After this, Otsu’s method was used for thresholding by implementing the `skimage.filters.threshold_multiotsu` function in the `scikit-image` package²⁷ using `nbins=3` parameter.

b. If just using EPCAM as a surface marker in segmentation, does this affect the quality of segmentation for immune and mesenchymal cells?

This question allows us to elaborate on our segmentation approach. Our pilot tests on this tissue informed our decision to have a segmentation strategy making use of both DAPI staining (as a nuclear marker) and the EPCAM antibody staining (as membrane marker). In fact, as illustrated

in the figure above (which is a smaller selection of our revised **Supplementary Figure 3**), the StarDist¹⁰ nuclei segmentation, relying solely on the DAPI staining, proved

¹⁰ Weigert, M., Schmidt, U., Haase, R., Sugawara, K. & Myers, G. in 2020 IEEE Winter Conference on Applications of Computer Vision (WACV). 3655-3662 (2020).

reasonably effective for segmenting cell types like endothelial, immune and mesenchymal cells, thanks to their sufficiently spaced nuclei enabling accurate segmentation. However, our pilot tests revealed difficulties in accurately segmenting epithelial cells within airway structures, where nuclei are densely packed. To overcome this, we incorporated EPCAM in our antibody panel, specifically to enhance the segmentation of epithelial cells. Consequently, the absence of EPCAM expression on non-epithelial cells does not detract from the segmentation algorithm's ability to segment these cells, which were effectively distinguished using nuclear segmentation alone.

In our initial submission, former **Supplementary Figure 2** (now **Supplementary Figure 3**) was providing an insight into the performance of segmentation for the various main cell types at each developmental week, however, we recognized the need to expand this figure by providing a more zoomed-in versions of the original selection of images as the performance of segmentation for different cell types was not clearly visible. We revised the **Supplementary Figure 3** and we believe this figure now provides a more thorough overview of the fit of the segmentation mask for different cell types.

c. No details provided for how Perseus was used to exclude non-lung areas or artifacts.

We appreciate the comment regarding the selection of tissue areas for downstream analyses. In the initial developmental stages at weeks 6 and 8.5, due to the small size of the organ, non-lung embryonic tissues were inadvertently included with the lung tissue, necessitating their exclusion. Then in later stages, with larger tissue samples available, we encountered staining artifacts particularly at the tissue peripheries. Including these peripheral regions in our downstream analyses led to clusters that misrepresented the underlying biology; hence, we opted to exclude these areas to maintain data integrity. Former **Supplementary Figure 3A** (now **Supplementary Figure 4A**) in our initial submission demonstrated this process, showing the polygons we drew in Perseus for each sample, which provides a transparent view of our criteria for distinguishing non-lung areas and peripheral staining artifacts. Additionally, **Supplementary Figure 4B** detailed the number of cell counts obtained after each data processing step, including area selection via Perseus. We revised **Supplementary Figure 4**, shown also below and its legend so that these details can be more clearly seen and understood.

d. Comments on how the excluded regions would affect results would be helpful. Are there any cell populations that would be missing from certain weeks because of cropping?

This is an excellent comment. As discussed in response to the previous point, excluding peripheral staining artifacts was necessary, but did indeed impact the representation of specifically mesothelial cells and chondroblasts. Mesothelial cells, present at all developmental stages as shown in **Figure 1D**, were inadvertently removed from downstream analyses starting from week 8.5 due to the exclusion of peripheral areas. Similarly, chondroblasts, which were present at week 8.5, were also excluded due to the initial area selection. To address this issue for chondroblasts, we adjusted our selection in Perseus to encompass a slightly larger area for the week 8.5 sample (as shown in the updated version of **Supplementary Figure 4A**) and performed all the downstream analysis from scratch. This adjustment allowed us to clearly identify and successfully include chondroblasts in our analysis. However, we were unable to rectify the exclusion of mesothelial cells in the downstream analyses of later timepoints.

We acknowledge that this important detail was not discussed in our initial submission. We have now reported this limitation in the Results section of the revised manuscript to provide clarity on the potential impact of our area selection on the results.

Page 5, line 26:

*“As discussed in the **Supplementary Information**, a few cell type clusters depicted in **Figure 2A-B** were not consistently represented across all weeks due to a number of technical reasons: Uneven signal intensities in the peripheral regions of the tissue sections necessitated the exclusion of these regions from downstream data analysis. Consequently, mesothelial cell type clusters were absent from 8.5 weeks onwards (**Supplementary Figure 4A-B**), despite their presence in all weeks as shown in **Figure 1D**.”*

We also added a **Supplementary Discussion Point** to Supplementary Information file where we explained the reasons for why a number of other cell type clusters were not represented uniformly across all the weeks.

e. Example code would help facilitate interpretation of analyses and processing steps.

We provided example code for all the steps of our analysis pipeline in the following Github repository, together with the segmentation outputs in .csv file format for each week which can be used following the instructions in the provided example code. We added now a Code Availability section in the revised manuscript:

Page 27, line 17:

“Segmentation of fluorescence images was performed using PIPEX: <https://github.com/CellProfiling/pipex>, archived in Zenodo (10.5281/zenodo.11642375). The code for all programmatic analysis performed in this study is available in the following GitHub repository: <https://github.com/CellProfiling/HDCA-FetalLung-SpatialProteomics/>, archived in Zenodo (10.5281/zenodo.11650173)”

Minor points

1. Visualization.

a. It is often difficult to interpret relative color intensities on the heatmaps or network diagrams as evidence for various claims throughout the manuscript. Where possible providing figures directly comparing co-localization Z-scores or cell distances would clarify the results. This would relate to claims such as the consistent colocalization of epithelium and ASM cells from week 8.5 on (Figure 3B), where from the heatmaps it only appears to be colocalized in 8.5 and 12.

As previously noted in our response to a related comment, we provided line plots summarizing the temporal changes in the scaled neighborhood cell type enrichment Z-scores among the various cell types (**Supplementary Figure 14**). We then also merged the different timepoints for each main cell type to compare the degree of homotypic neighboring patterns for proliferating and non-proliferating cells (**Supplementary Figure 18**). We believe these changes now allow for a more straightforward comparison of the spatiotemporal trends observed across the different weeks.

In addition it appears that the color scale for week 8.5 in figure 3B has a 0 value which is in the red area which could be misleading, although this could be a cropping issue.

This was indeed an error introduced when assembling the initial version of Figure 3B: the color legend axis values for week 8.5 got vertically misaligned compared to the color legend itself. In the revised version, the enrichment Z-scores have been scaled and merged across all five weeks and there is now only one color legend for the scaled enrichment Z-scores.

b. The color palette for figure 6E makes it difficult to distinguish different cell colocalizations.

We thank the reviewer for their observations regarding the color palette in **Figure 6E**. We initially employed a color scheme that aimed to consistently represent main cell types using primary colors and their subtypes using shades. During our internal reviews, we debated several times between this approach, or adopting a varied color palette without strict adherence to subtype color coding. Initially favoring the former for thematic consistency, we've now reconsidered based on the feedback, as this choice compromised the clarity needed to distinguish between different cell subtypes effectively. We have revised the color palette for **Figure 6E**, and for several other figures. We hope these changes alleviate the difficulties previously encountered in interpreting the data and enhance the presentation of our results.

c. There are no error bars in figure S10C.

We thank for highlighting the absence of error bars in this figure. We acknowledge that this was an oversight in our initial submission. To address this, we have now added error

bars, which indicate the standard error of the mean, to both former **Supplementary Figure 10C** (now **Supplementary Figure 18C**, shown below) and the related **Figure 5F** to more accurately represent the variability within the percentages in selected regions.

2. Figure order

a. Some of the supplementary figures are out of order with their position in the text.

We appreciate the reviewer's attention to the order of the supplementary figures in relation to the text. We believe that the real issue was that the Methods section, where several of the supplementary figures were already referred to, preceded the Results section in our initial submission. This placement may have given the impression of a discrepancy in figure order when reading the Results section. To enhance continuity, we have revised the manuscript by relocating the Methods section to follow the Results section. We hope this change ensures a better alignment of the supplementary figures with their corresponding text references. And we are grateful for the reviewer's comment, which prompted this beneficial restructuring.

Reviewer #2 (Remarks to the Author):

This well-written manuscript investigated the spatial organisation of human foetal lung cell types between 6-13 post conception weeks at protein level through high-parametric microscopy. An oligo-barcoded antibody panel (30 antibodies) representative of the main foetal lung cell types during the first trimester of gestation was designed and 1 million cells were analysed at single-cell resolution. First, different cell types, their main expressed markers and their abundancy across developmental stages was determined. Next, the authors studied the adjacency patterns among the identified cell types, which highlighted that while cells of the same type predominantly shared adjacency, this pattern was dynamic throughout developmental time points. Furthermore, they examined cellular interactions on a higher order by looking into multi-cellular units, suggesting an early emergence of lung tissue organisation, and highlighting the importance of spatial

organisation of cells and their interactions in orchestrating lung development. They also explored the composition of the different proliferative cell types across developmental stages, and their adjacency patterns, indicating a temporal sequence in lung development, as well as a spatial gradient of proliferation along the proximal-distal axis of the developing epithelium. Lastly, they delved into the immune landscape, reporting on the abundancy of various immune cells across developmental timepoints, their spatial proximity patterns with other immune cells, and their spatial arrangement patterns within the tissue.

This would be an extremely valuable resource to have, but based on the following comments, I do not think it is ready for publication in its current form.

We thank Reviewer 2 for their constructive feedback on our manuscript and for acknowledging the potential value of our work to the field. We appreciate the constructive criticisms and we tried to address these now in the revised version of our work. Please see our point-by-point responses and the changes made to the manuscript below.

MAJOR POINTS

1) The number of biological replicates is only 1 per 5 timepoints. One time-point in the embryonic and 4 in the pseudoglandular stages with no mention of technical replicates. This is clearly insufficient.

The procurement of human embryonic organs, which offer a limited number of sections to begin with, and which must be maintained in fully intact conditions for spatial analyses, presents substantial challenges. This requirement contrasts with methods that utilize dissociated tissues, as our approach depends on the *full* integrity of the whole organs, without any physical defects such as partial detachments. Despite these stringent requirements, the data we derived from the carefully selected samples - each chosen for its exemplary preservation post-extraction - still offer a unique and invaluable resource to study the human embryonic lung development.

We recognize the importance of including multiple biological replicates and wish we could access many more intact human embryonic lung specimens suited for this study, in addition to dozens of sections we have already used for multi-step validation of over 50 antibodies on these tissues to create our antibody panel from scratch. Until there are more extensive global initiatives for the collection and biobanking of human embryonic organs, we hope that the reviewers and future readers of the work will appreciate the challenges involved in studying such rare samples and recognize the scientific value that even limited data can provide to the developmental biology community. Recognizing the

constraints imposed by the available sample size, we have incorporated a discussion point into the manuscript:

Page 15, line 12:

“Second, our study is based on the analysis of lung samples from embryos/fetuses aged 5-13 post-conception weeks, sourced from one donor at each developmental time point, covering five timepoints in total. Analyzing more timepoints would provide a higher resolution of the temporal changes in lung development. [...] Larger, more uniform, and independent cohorts of tissue sections are needed to yield sufficient statistical power to confirm the protein expression and relationship patterns, as well as their temporal changes, described in our study.”

2) I cannot see any statistical analyses done.

Our initial submission provided a statistical comparison for the fraction of proliferating cells in small SOX2^{high}, large SOX2^{high} and SOX9^{high} airway regions in individual weeks of 11 to 13 (**Supplementary Figure 19**), as well as cumulatively across these weeks (**Figure 5F**). Then, the cell type adjacency analysis in our initial submission also was based on the use of neighborhood enrichment statistics reporting Z-scores, instead of counts. These were giving an overview of cell type adjacency patterns in individual weeks but were missing to provide an overall view across the weeks. To address this, in the revised version, we scaled the Z-scores in and across all weeks to provide a more straightforward comparison of changes in cell type adjacency patterns across the weeks. To support our finding that non-proliferating cells have a more homotypic cell type adjacency pattern compared to their proliferating counterparts, we provided a statistical analysis now in our revised manuscript (**Supplementary Figure 18**). In the revised manuscript, we also provided a statistical comparison for the marker expression patterns in artery-distant versus artery-close immune cells (**Figure 7C**). The related Methods sections have been revised. We believe the meaningful statistical comparisons are now provided in the revised version of our manuscript.

3) The paper is very descriptive with few if any novel findings.

We value the reviewer's perspective and welcome the opportunity to elucidate the novel contributions of our study. While the core of this work is explorative and descriptive, its novelty remains uncompromised. To the best of our knowledge, our study presents the first-ever resource offering single-cell resolved protein-level expression data to elucidate the composition and spatial relationships of key cell types in the human embryonic lung

during the first trimester. To our knowledge, no prior publication has employed such a comprehensive multiplexed antibody-based approach for spatial analysis in this context.

Three recent, mainly single-cell transcriptomics-based studies^{11,12,13}, including one involving co-authors of this manuscript, have explored human lung tissue at a similar developmental stage, none have conducted a single-cell resolved, protein-level analysis utilizing a diverse set of antibodies. Our research fills this critical gap by providing an extensive application of this method to investigate developing human lung tissue.

While sc-RNAseq offers single-cell resolution and transcriptome-level information, it lacks the entire spatial context in tissues, and current spatial transcriptomics approaches sacrifice single-cell resolution for spatial information. Our single-cell resolved, protein-level spatial assay provides unique advantages *e.g.* for mapping the continuous network of vasculature in whole tissue sections or characterizing the spatial neighborhood of individual immune cells or studying the degree of proliferation across cell types and tissue regions of interest.

In light of this explanation, we would like to highlight some aspects and findings of our study underscoring its novelty and value:

- It offers whole tissue level architecture of the developing human lung by providing protein-level snapshots of how the spatial network of immune cells, endothelium and lymphatic structures expand throughout early human lung development (**Figure 2D-E**)
- It offers a protein- and single cell-level composition of the cell types present in the developing human lung and how this compares to the single cell transcriptomics-based cell type composition (**Figure 2C**)
- It offers a quantitative overview and comparison of neighboring patterns among the different main cell types present in the developing human lung (**Figure 3B, Figure 4**)
- It describes a spatial gradient of proliferation along the proximal-distal axis at protein level (**Figure 5F**)
- It offers a comparative and quantitative overview of the degree of proliferation in cell types and subtypes, in particular immune cells (**Figure 5B-C, Figure 6D**)

¹¹ Sountoulidis, A. *et al.* A topographic atlas defines developmental origins of cell heterogeneity in the human embryonic lung. *Nature Cell Biology* 25, 351-365, doi:10.1038/s41556-022-01064-x (2023).

¹² He, P. *et al.* A human fetal lung cell atlas uncovers proximal-distal gradients of differentiation and key regulators of epithelial fates. *Cell* 185, 4841-4860.e4825, doi:10.1016/j.cell.2022.11.005 (2022).

¹³ Barnes, J. L. *et al.* Early human lung immune cell development and its role in epithelial cell fate. *Science Immunology* 8, eadf9988, doi:10.1126/sciimmunol.adf9988 (2023).

- It pinpoints that human lung macrophages become MRC1+ (CD206+) between post conceptional week 6 and 8.5 (**Figure 6A**) and it also describes a special radial arrangement of immune cells around the arteries suggesting physical interactions between them and the developing vasculature, thus their possible role in vascular remodeling, which has been reported in mice but not in human (**Figure 7**)

While these insights may not reveal any completely unforeseen aspects of human lung development, or not have any directly translational significance for human diseases, they still represent important pieces of information increasing our current understanding of human lung development. We also would like to remind that we have curated and deposited the imaging data and cell type annotations described in this manuscript into an interactive and publicly accessible portal. This portal offers additional single-cell and spatial transcriptomics datasets from a partially overlapping cohort of human embryonic lung samples, providing a comprehensive resource for researchers. We believe the resultant data serves the scientific community at large and it will be instrumental for researchers in the field.

In closing, the explorative nature of our work does not diminish its novelty but rather reinforces its value as a steppingstone toward an improved, protein-level understanding of the developing human lung. We believe that the data, the presented antibody resource, and observations reported here will catalyze future research that will unearth new biology.

4) I acknowledge the effort used to validate the 30 markers, but nevertheless it is very difficult to reliably identify granular subtypes using only 30 markers across epithelial, mesenchymal and immune compartments.

We fully agree with the reviewer that using 30 protein markers cannot comprehensively detail all cell types and granular states in the developing human lung, and this was not our intention. Building assays like this is incredibly laborious, compounded by the limited availability of suitable antibodies for conjugation. Importantly, we evaluated a total of 72 antibodies to establish the final 30-plex antibody panel (detailed in **Supplementary Table 1**). For comparison, the company Akoya Biosystems, commercially operating for over seven years and providing our high-parametric imaging platform, currently offers pre-conjugated and validated antibodies for only 23 unique targets for fresh-frozen human tissue.

Our 30-plex panel was built meticulously through a rigorous quality control (QC) process spanning over two years and involving several stages:

1. Test of primary antibody performance on tissue

2. If Stage 1 is successful, barcode conjugation
3. If Stage 2 is successful, test of conjugated antibody performance
4. If Stage 3 is successful, final test of conjugated antibody as part of panel

Due to constraints on the number of additional antibodies we could test per target, some cell types are not represented in this dataset. We prioritized having a reliable set of markers to assign each cell to a main cell type, essential for downstream analyses, particularly spatial analyses, which rely on resolving the main cell type composition of the entire tissue. Our intention was also to build a panel that allowed exploration of aspects which single-cell sequencing and spatial transcriptomics studies have left partially unexplored, such as the spatial composition of immune cell subtypes and proliferative cell states.

Looking ahead, we would love to expand this panel, but we do believe this is beyond the scope of the current study. Our manuscript now also provides the scientific community with valuable insights into antibody performance on this specific and relatively unexplored tissue type. This enables researchers to construct much larger panels with far less expenditure of time and resources than what was required of us. We also aspire to see the initiation of more extensive and global initiatives aimed at the collection and biobanking of human embryonic organs. Broad antibody testing endeavors on these understudied human tissues, derived from small-scale organs, critically depends on securing substantial collections of sample cohorts.

We agree though with the reviewer that this is a limitation of our approach. We now added a brief discussion point regarding this limitation of the study:

Page 15, line 9:

“Our study has a few limitations. First, although our 30-plex antibody panel successfully identified the main cell type compositions in epithelial, mesenchymal, and immune compartments, it offered limited depth of analysis for certain key cell types such as secretory or neuroendocrine cells.”

MINOR POINTS

- 1) Why are some cell types missing, e.g. a) mesothelial cells only present at 6pcw and then not, b) neurons only detected at 11pcw?

This is a very good comment, and we acknowledge that it was an oversight not to explain and discuss this important detail more clearly in our initial submission.

As shown by the podoplanin (PDPN) antibody staining patterns in **Figure 1D**, mesothelial cells were indeed present at all five developmental weeks. However, as detailed in **Supplementary Figure 4A**, in the relatively larger tissues of the 8.5 week and all later-stage lung samples, we experienced uneven signal intensity for various markers in peripheral regions, causing issues in downstream clustering after image segmentation. Thus, we removed these peripheral regions before downstream data analysis. This impacted the representation of mesothelial cells in all weeks except the week 6 in the downstream analyses.

Similarly, chondroblasts, which were present at week 8.5, were also excluded due to the initial area selection. To address this issue for the 8.5-week-old sample, we now adjusted our selection in Perseus to encompass a slightly larger area (as shown in the updated version of **Supplementary Figure 4A**) and re-analyzed the dataset. This adjustment allowed us to clearly identify and show a chondroblast cluster in our analysis for the 8.5-week-old sample (**Figure 2B**). Chondroblasts were also missing in the 12-week-old sample, but for a different reason: Due to limited tissue availability, the section presented for the 12-week-old sample originates from a relatively more peripheral region of the organ compared to other samples. Considering that the analyzed tissue section contains no distal parts of the airway network, it is expected not to detect chondroblasts.

Regarding neuronal cells, we re-examined our images and, using a combination of markers we identified them more consistently across the weeks from 8.5 weeks onwards. We therefore performed clustering again and annotated the clusters spatially corresponding to neuronal cells from weeks 8.5 onwards.

For the pericytes and the airway fibroblasts, we did not have single specific markers for these cell types. Instead, like neuronal cells, we used a combination of several markers that provided clusters with characteristic signatures aligning with the positioning and morphology of these cells in the tissue images. This approach, while demonstrating the strength of multi-parametric image analysis at the protein level, is prone to underperform if one or more of the markers do not perform equally well (e.g. ACTA2 in week 12 compared to weeks 11 and 13), affecting the clustering performance.

Adventitial fibroblasts are localized within the bronchovascular bundles of the lung¹⁴. Among the analyzed sections, only those from the week 11 sample contained such a central part of the organ. This provides an explanation about their absence in more peripheral lung sections.

¹⁴ Sountoulidis, A. *et al.* A topographic atlas defines developmental origins of cell heterogeneity in the human embryonic lung. *Nature Cell Biology* 25, 351-365, doi:10.1038/s41556-022-01064-x (2023).

We now added this response as a **Supplementary Discussion Point** into the Supplementary Information file, and also added a shorter version of this in the Results section:

Page 5, line 26:

*“As discussed in the **Supplementary Information**, a few cell type clusters depicted in **Figure 2A-B** were not consistently represented across all weeks due to a number of technical reasons: Uneven signal intensities in the peripheral regions of the tissue sections necessitated the exclusion of these regions from downstream data analysis. Consequently, mesothelial cell type clusters were absent from 8.5 weeks onwards (**Supplementary Figure 4A-B**), despite their presence in all weeks as shown in **Figure 1D**. Chondroblast clusters were also absent in the 12-week-old lung sample as the section presented for the 12-week-old sample originated from a relatively more distal region of the organ compared to other samples. Similarly, adventitial fibroblast clusters, shown to localize within the bronchovascular bundles of the lung⁷, were identified only in the 11-week-old sample originating from a central section of the organ, and were absent in the 12- and 13-week-old human lung samples.”*

We also added a new limitations paragraph into our revised Discussion and briefly mentioned this:

Page 15, line 15:

“Additionally, the lung tissue sections from different developmental weeks did not offer an equally balanced representation of the distal or peripheral microanatomy, leading to inconsistent identification of certain cell types, such as chondroblasts and adventitial fibroblasts, in our downstream analysis. Larger, more uniform, and independent cohorts of tissue sections are needed to yield sufficient statistical power to confirm the protein expression and relationship patterns, as well as their temporal changes, described in our study.”

2) ‘Dendritic cells, NK cells and B cells emerged from 11 weeks onwards (Figure 6B) emphasizing the distinct timelines of immune cell formation during development.’ - In Barnes et al., Sci Immuno 2023 paper, all of these cell types were present from 8 weeks according to scRNAseq; plus immunohistochemistry shows DCs at 8 pcw.

As far as we can conclude, the Barnes *et al.* 2023¹⁵ study successfully identified dendritic cells (DC2's) using CD1C as a marker in IHC. In our panel, we did not have such a marker for DCs, but we were able to annotate them in later weeks based on HLA-DR expression and lack of other lineage markers. For the 8.5-week-old sample, we followed the same approach but we could not identify a separate cluster of HLA-DR⁺ cells. Based on this comment, we now revisited both the images and the segmented dataset (by manual gating) for the 8.5-week-old sample, however, we could again not identify only HLA-DR⁺ cells; they were also expressing MRC1, as indicated in the dotplot below (cluster marked in red). We therefore cannot report their presence.

Recent scRNA-seq studies reported that “B, T, and NK cells becoming prominent from 15 pcw”, citing the findings of He *et al.* (2022)¹⁶. The slightly more recent scRNA-seq study by Barnes *et al.* 2023¹⁷ indeed reports their presence in earlier 8 and 9 weeks and citing their finding they “confirmed the presence of T, NK and B cell types by IHC in tissue sections at 12 and 20 pcw”, and not in earlier weeks. Our data reports the presence of NK and B cells from 11 weeks onwards at protein level using IHC. The reason that these two cell types were not identified in earlier weeks using IHC could be due to the random histologic sampling in IHC (as compared to dissociation of the whole organ and enrichment of immune cells prior to sequencing) and/or discrepancy between RNA and protein expression levels for marker genes for these cell types.

In the revision, we commented about these two points in the Discussion as follows:

Page 14, line 14:

“Previous transcriptomics-based studies reported that these cells become prominent first after week 15⁶. Here, we show the presence of NK and B-cells in

¹⁵ Barnes, J. L. *et al.* Early human lung immune cell development and its role in epithelial cell fate. *Science Immunology* 8, eadf9988, doi:10.1126/sciimmunol.adf9988 (2023).

¹⁶ He, P. *et al.* A human fetal lung cell atlas uncovers proximal-distal gradients of differentiation and key regulators of epithelial fates. *Cell* 185, 4841-4860.e4825, doi:10.1016/j.cell.2022.11.005 (2022)

¹⁷ Barnes, J. L. *et al.* Early human lung immune cell development and its role in epithelial cell fate. *Science Immunology* 8, eadf9988, doi:10.1126/sciimmunol.adf9988 (2023).

week 11 onwards using immunofluorescence. A more recent study¹⁰ reported the presence of dendritic cells from week 8 onwards using the marker CD1C for immunofluorescence. In our panel, we did not have such a marker for dendritic cells, but we were able to annotate them in later weeks based on HLA-DR expression and lack of other lineage markers, thus we report the presence of dendritic cells from week 11 onwards, which does not exclude the possibility that they are present in earlier weeks.”

3) Overlapping numbers on UMAP in Fig S5A.

We appreciate the reviewer’s attention to this detail. We realize that the numbers in the UMAP plots were not informative, we now provided color coding in this specific UMAP and all other UMAP figures in the revised version of our manuscript.

4) Figure 6, no CD8 T-cells identified.

As detailed in **Supplementary Table 1**, our study tested an extensive panel of 72 antibodies, yet the data presented in the manuscript are derived from 30 of these 72 antibodies that met our criteria for staining performance and reliability. Among the initial panel was indeed an antibody against CD8, alongside others intended for further phenotyping of T-cells. Regrettably, the CD8 antibody did not demonstrate sufficient staining quality on this embryonic human lung cohort to warrant its inclusion in the downstream analysis. Consequently, we opted to annotate ILCs and T-cells together, without further classification.

Validating such a comprehensive antibody panel on a relatively unexplored tissue type, especially one sourced from a small-sized organ yielding a limited number of sections, presents considerable challenges, necessitating some degree of prioritization and compromise. Our goal was to reveal the spatial composition of as many major cell types as possible with the highest data quality. Although a more detailed characterization of several cell types, including T-cells, would undoubtedly be valuable, the limitations imposed by donor and tissue availability limited us regarding how much additional effort and resources we could put on every single failed antibody. We hope to include successfully validated antibodies in future iterations of our research, allowing for a more granular exploration of immune cell populations within the human embryonic lung. We believe that the groundwork laid by the current dataset provides a good foundation for such subsequent studies.

5) Why was 50microns chosen as the radius to use for looking at immune cells around arterial areas. This seems quite a large distance.

We now added a **Supplementary Figure 2B**, shown to the right, to illustrate the reasoning behind the selection of a radius of 50 μm around each detected immune cell around arteries. As shown also here, instead of analyzing the neighborhood composition of these immune cells across only one or two layers of cells, we wanted to do it over five-six layers of cells so that cells, such as the one pinpointed at the center of the 50 μm radius in this figure, can still be considered as an “arterial” immune cell, as it is still in close proximity to the arterial area. We acknowledge that it is still an arbitrary choice of distance, but we believe it can be considered an empirically acceptable selection given the exemplified snapshot and the explanation.

6) Fig 4C – it is hard to differentiate between some of the lines on this plot, especially where colours are so similar e.g. week 11 plot.

We now changed the tool for the unsupervised identification of spatial domains and opted also to remove this former Figure 4C, where the quantitative dispersion metrics (Ripley's L) were visualized for each domain. We believe such an analysis with domains generally containing more mixed cell types was not relevant anymore.

7) Figure 5A and B – IHC images in 5A show only a fraction of the results presented in 5B. It is important to include all IHC images that support results for the composition of proliferative cells shown (e.g. Mesenchymal, SOX2^{high} epithelium, SOX9^{high} epithelium, lymphatic endothelium, ASM, VSM), at least somewhere in the supplementary information.

To more clearly demonstrate the proliferation across various cell types directly within the images, we have prepared a new figure where we provide snapshots of overlaid expression of Ki67 and one or two respective markers for each main cell type. Because this figure consisted of a large number of images, we opted to add this as a supplementary figure, **Supplementary Figure 17**. The figure below shows a smaller portion of the new **Supplementary Figure 17**.

8) General comment. The colours are too similar to be able to differentiate between the cell types in these plots, such as Figure 6C, so the colours need changing, to allow this. If that doesn't help, then the plots should be removed, as you can't discern any useful information from them. This also applies to Supplementary Figure 7 (too many similar shades of blue), Figure g (B,D,E), Figure 5B (cannot distinguish between Mesenchymal and ASM cells).

We appreciate the reviewer's critique concerning the color schemes in our figures. As previously noted in our response to a similar observation by Reviewer 1, our original intent was to employ primary colors for major cell types and shades of these colors for their subtypes. While this approach was designed for thematic cohesion, it has become clear that it does not sufficiently differentiate related cell subtypes. Based on your valuable feedback, we've now revised the color schemes in all main figures (and all related supplementary figures), opting for a more distinct and varied color palette. We hope these modifications enhance the clarity and interpretability of the visual data presented.

Reviewer #3 (Remarks to the Author):

The manuscript by Sariyar et al. aims to unravel the cellular heterogeneity and cellular protein landscape in the developing respiratory system by exploring the cell phenotype in the lungs,

trachea, and vasculature. As part of a Human Developmental 7 Cell Atlas (HDCA) initiative, the authors describe a spatiotemporal organization of lung cells at different time points of development during the first trimester by applying a high-parametric tissue imaging approach using a 30-plex antibody panel. Using this novel technology, the authors define cell phenotypes of individual lung cells across five developmental timepoints in the first trimester, with a comprehensive spatially resolved cell type composition, rate of cell proliferation and cellular spatial location. These data are very insightful and provide an excellent resource for studies investigating development of the respiratory immune system in prenatal and early life. Furthermore, this manuscript could serve as an essential platform for interrogating human respiratory health and disease. Overall, the data provided is very impressively carried out with some sections needing further clarity.

We thank Reviewer 3 for their insightful and constructive feedback on our manuscript. We appreciate the recognition of our work's contribution to the field and have now addressed the areas needing further clarity as suggested by the reviewer. Please see our point-by-point responses and the changes made to the manuscript below.

Some of the points highted are below:

Major points:

1. The authors use DAPI as a nuclear marker and EPCAM as a membrane marker however EPCAM is not expressed on non-epithelial cells? So how confident can the authors be with segmenting all the lung cells accurately?

As also addressed in the response to a similar query by Reviewer 1, our segmentation strategy made use of both DAPI staining (as a nuclear marker) and the EPCAM antibody staining (as membrane marker). We used DAPI staining for nuclear identification across all cell types and incorporated EPCAM staining to

specifically enhance the segmentation of epithelial cells, given its membrane localization in these cells. We understand the concern regarding EPCAM's expression limited to epithelial cells and its implications for accurately segmenting non-epithelial cells. It's important to reiterate that the StarDist nuclei segmentation technique, which primarily employs DAPI staining, was adequately effective for the segmentation of endothelial,

immune, and mesenchymal cells. These cell types, characterized by less densely packed nuclei, did not necessarily require membrane staining for their segmentation (as elaborated in our earlier response and demonstrated in the figure above, which is a smaller selection of our revised **Supplementary Figure 3**). For epithelial cells, particularly within airway structures where nuclei were densely clustered, the addition of EPCAM significantly improved segmentation. Therefore, our confidence in the segmentation of all lung cell types, including non-epithelial cells, rests on the robustness of the StarDist technique supplemented by EPCAM for epithelial cells, ensuring comprehensive and accurate cell identification across the lung tissue.

2. The authors describe Sox9^{high} and Sox2^{high} epithelial cells but do not show any marker trajectories for ciliated, secretory and neuroendocrine cells. This would be important to understand how physiological epithelial differentiation and turnover occurs during development.

We fully agree with the reviewer that these three epithelial cell types are important. As listed in **Supplementary Table 1**, we tested a total of 72 antibodies in this study, and the data described in the manuscript originates from only 30 of those 72 antibodies. Among the set of 72 antibodies, we had one antibody targeting the protein ASCL1, which is important for pulmonary neuroendocrine cell development. It included another antibody targeting the surfactant protein SFTPC, produced and secreted by type II alveolar cells. These, and many other antibodies listed in **Supplementary Table 1**, which we wanted to include in our panel to optimistically target several other more specialized cell types, did unfortunately not pass our initial antibody screening consisting of several stages:

1. Test of primary antibody performance on tissue
2. If Stage 1 is successful, barcode conjugation
3. If Stage 2 is successful, test of conjugated antibody performance
4. If Stage 3 is successful, final test of conjugated antibody as part of panel

For the failed antibodies, we were limited in the number of additional different antibodies we could test per target, which resulted in certain cell types not being represented in this dataset.

Ciliated epithelial cells are also not represented in our dataset. The scRNA-seq dataset published in the related previous work by Sountoulidis *et al*, 2023¹⁸ revealed a significantly small fraction of ciliated cells (indicated with a black arrow in the figure below) in a combined analysis of approx. 163,000 cells from human lungs at 5 to 14 post-conceptual weeks of development. The suggested ciliated epithelial cell marker from this study, FOXJ1, was therefore deprioritized, as from a technical point of view, it was

¹⁸ Sountoulidis, A. *et al*. A topographic atlas defines developmental origins of cell heterogeneity in the human embryonic lung. *Nature Cell Biology* 25, 351-365, doi:10.1038/s41556-022-01064-x (2023).

more pressing for us to be able to annotate each and every cell accurately as belonging to a main cell type with the limited number of markers we could have to enable any downstream spatial analyses.

We believe now our panel will serve as a helpful starting point for many other researchers to construct much larger and detailed antibody panels with far less expenditure of time and resources than what was required of us. Nevertheless, we now added a brief discussion point regarding this limitation of our study:

Page 15, line 9:

“Our study has a few limitations. First, although our 30-plex antibody panel successfully identified the main cell type compositions in epithelial, mesenchymal, and immune compartments, it offered limited depth of analysis for certain key cell types such as secretory or neuroendocrine cells.”

3. The authors analyse each condition separately and extract data out to perform statistical analysis between conditions. This also explains why there are so many graphs generated as they would need one set of graphs for every condition. In this case it is quite difficult to make strong conclusions purely because you can't statistically compare one network graph to another, which at the moment as I understand are selected based on visual indication.

For example, in figure 3 the authors have all of their correlation matrices, where you can see some small differences perhaps, but these graphs do not show a statistical significance between the weeks so I don't think you could say there are significant difference between the matrices. One would have to pull out the individual Z scores, plot them and do statistical testing of those to get anything definitive.

This was a valid criticism made also by Reviewer 1. To address this in the revised version, we first scaled the neighborhood enrichment Z-scores obtained in each week in order to provide a more comparable overview of the cell-cell adjacency patterns across the different weeks (**Figure 3B**). In addition, for the main cell types commonly identified and annotated across all weeks, we provided visuals summarizing the temporal changes in

their scaled enrichment Z-scores for neighboring their own or the rest of the other cell types (**Supplementary Figure 14**). We then also merged the different timepoints for each main cell type to compare the degree of homotypic neighboring patterns for proliferating and non-proliferating cells (**Supplementary Figure 18**). We believe these changes now allow for a more straightforward comparison of the spatiotemporal trends observed across the different weeks.

4. Also it is unclear how many donor tissues were obtained per developmental time point and whether the clinical status of the donating mother was assessed. These would strongly impact lung developmental cell states.

We thank the reviewer for this comment, as we realize that these two details were not explicitly stated in the manuscript: The data presented in the manuscript originates from one donor per developmental time point. The women donating these embryos were self-reported as “healthy”, and the reason to terminate their pregnancies was not medical. To protect personal integrity, no further details on the health status of the pregnant women were disclosed to the researchers. None of the donated embryos were suspected of any malformations, chromosomal aberrations etc.

We now added these details to the “Prenatal tissue samples and ethics” section under Methods:

Page 16, line 3:

“The lung samples from embryos/fetuses aged 5-13 post-conception weeks (pcw) were sourced from one donor at each developmental time point. These were obtained after elective abortions, with both oral and written consent from the donors who were informed of their rights to withdraw consent at any time. The recruitments were done by midwives uninvolved in the conducted research. Inclusion criteria included donors being 18 years of age or older and fluent in Swedish. Exclusion criteria excluded abortions performed for medical reasons, socially compromised donors, or any indications of uninformed consent. The donors were self-reported as “healthy”. None of the embryos were suspected of any malformations and chromosomal aberrations.”

Authors' Response to Reviewer Comments

We are grateful to all three reviewers for their reevaluation of our revised manuscript. We now made further adjustments to the manuscript based on the additional feedback. Below, we provide a point-by-point response to their questions and comments.

Reviewer #1 (Remarks to the Author):

I commend the authors for doing many analyses to support and strengthen the manuscript, and most of my comments have been addressed. I particularly find the line plots for colocalization, the comparison to scRNA-seq, and a more clear comparison of proliferation differences to be great additions. I also believe their discussion of limitations and nuanced technical aspects to be greatly informative to the reader when assessing the data.

We thank Reviewer #1 for their excellent guidance in suggesting all these concrete additions and changes to our manuscript, including those addressed in this round. Their input highlights the incredibly valuable role that peer reviewers play in scientific publishing.

Minor comments:

- The claim that neuronal cells had increased adjacency with ASM cells from week 8.5 on doesn't appear to make sense because I don't see neurons identified in week 6.

We appreciate the reviewer's feedback and apologize for the confusion caused by a wording mistake. Our intention was not to imply that neuronal cells were also identified at week 6, but rather to state that the neuronal cells, identified in our dataset from week 8.5 onwards, exhibited increased adjacency with ASM cells as compared to their adjacency patterns with other cell types. We corrected the related Results section accordingly and ensured there is no ambiguity regarding the presence of neuronal cells at different developmental stages.

Page 8, line 6:

“Additionally, neuronal cells identified in our dataset from week 8.5 onwards exhibited increased adjacency with airway smooth muscle cells compared to their adjacency patterns with other cell types (Figure 3B).”

- It is difficult to say the cell density is stable without replicates. It can be said it is generally consistent although increases slightly in later weeks

We agree, and we prefer the way the reviewer rephrased the observation regarding cellular density. We now changed this accordingly in the revised manuscript.

Page 5, line 10:

“Cellular density was generally consistent across the developmental weeks, with a slight increase observed in the later weeks (Supplementary Figure 4C).”

- The claim that the domains cluster based on type rather than time point is very difficult to assess given the lack of replicates and it is not clear that that is the case based on the correlation heatmap.

The referenced Supplementary Figure 16 (also provided below) presents a heatmap summarizing the correlations among individual spatial domain cell type compositions identified at each developmental week. This analysis primarily served as a sanity check to ensure the consistency of spatial domains—comprising multiple cell types—identified through the unsupervised domain analysis across different ages. Our interpretation of this dendrogram suggested that domains enriched with the same cell types across different weeks exhibit greater similarity to each other than domains enriched with different cell types within the same week. In other words, we observed that the segregation of domains in the heatmap was driven not by sample age but rather by the cell type composition within the identified spatial domains. Accordingly, the correlation heatmap predominantly reflects segregation by the type of cells dominating the spatial domains rather than the age of the samples. However, this conclusion is more readily applicable to domains rich in homogeneous cell types, such as SOX2^{high} epithelial-rich, SOX9^{high} epithelial-rich, chondroblast-rich, and ASM-rich domains. We acknowledge that drawing the same conclusion for the more heterogeneous domains—rich in mesenchymal, endothelial, and immune cell types—is more challenging. Furthermore, as the referee correctly pointed out, the limited sample size warrants a more cautious description and interpretation. We now clarified the related sentence in Methods section and also rephrased the description in Results as follows:

Page 25, line 2:

“(…) similarity between cell type compositions in spatial domains across different weeks was assessed by calculating the Pearson’s correlation coefficient (**Supplementary Figure 16**).”

Page 9, line 7:

“Hierarchical clustering of correlations across all individual spatial domains and developmental weeks confirmed that domains enriched for the same cell type across different developmental ages exhibited greater similarity to each other than those enriched for different cell types within the same developmental age (**Supplementary Figure 16**). Domains rich in airway smooth muscle cells, chondroblasts, and $SOX2^{high}$ or $SOX9^{high}$ epithelium clustered distinctly, indicating consistent spatial organization across different weeks of development. The more heterogeneous mesenchyme-, endothelium-, and immune-rich domains clustered together (**Supplementary Figure 16**), reinforcing previously described patterns of cell type adjacency among these cell types.

Reviewer #2 (Remarks to the Author):

Thank you for submitting a detailed response to my comments.

MAJOR POINTS

- I understand the difficulty in obtaining intact human fetal lungs, but using only 1 biological replicate per time points with 5 points in total and only 1 technical replicate per time point is simply not enough to give confidence in the results. I am also not sure how statistics was done on only 1 technical replicate at each time point. I appreciate all the hard work which has been done so far, but to me more biological replicates are needed before considering publication. At the very least more technical replicates are needed. Would a transfer to a methods journal be more appropriate?

We appreciate the reviewer's recognition of the effort involved in this study and the challenges associated with obtaining human fetal lung samples. While we understand the concern regarding the limited number of biological replicates, we would like to clarify and expand on several key points regarding the robustness and value of the dataset.

First, we have already acknowledged this limitation in the Discussion section and provided context on how it impacts the interpretation of our findings. Despite this, the dataset comprises nearly 2.5 million cells, making it one of the most comprehensive single-cell protein-level studies of human fetal lung tissue to date. The quantitative data presented is based on a high-dimensional 30-plex antibody panel, developed through qualitative evaluation of dozens of slides stained with these and additional antibodies from the same embryos. Our results also align well with findings from other studies using scRNA-Seq, VISIUM, and spatially resolved transcriptomics techniques (e.g., Sountoulidis *et al.* 2023¹ and He *et al.* 2022²). The consistent spatial distribution patterns of cell types across samples collected at closely spaced weekly intervals further reinforce the reliability of the dataset.

Regarding statistical analysis, while we did not aim to detect statistically significant changes across developmental weeks longitudinally, we employed appropriate statistical tests to assess specific trends. For example, we evaluated whether non-proliferating cells exhibited a more homotypic cell type adjacency pattern compared to proliferating cells and whether there were statistically significant differences in the fraction of proliferating cells across SOX2^{high} or SOX9^{high} airway regions. That said, we have been careful to

¹Sountoulidis, A. et al. A topographic atlas defines developmental origins of cell heterogeneity in the human embryonic lung. *Nature Cell Biology* 25, 351-365, doi:10.1038/s41556-022-01064-x (2023).

²He, P. et al. A human fetal lung cell atlas uncovers proximal-distal gradients of differentiation and key regulators of epithelial fates. *Cell* 185, 4841-4860.e4825 (2022). <https://doi.org/10.1016/j.cell.2022.11.005>

avoid overgeneralization, ensuring that the interpretations remain well within the study's scope.

Finally, while we fully recognize the importance of biological and technical replicates, we believe the primary contribution of this work lies in providing a resource for the scientific community. This dataset offers access to spatially resolved protein-level data from human fetal lungs during the first trimester, and its public availability will facilitate further exploration and validation by other researchers.

In light of these points, we believe that this study, despite its limitations, is well-suited for a broad audience, such as the readership of Nature Communications. A transfer to a methods journal would not fully reflect its relevance as a resource developed under the Human Cell Atlas initiative.

MINOR POINTS

- In the Methods section, no karyotype analysis was done. This would be very important in view of only 1 sample being used per time point.

We thank the reviewer for raising this point. While karyotyping would not exclude the presence of point mutations, small deletions, or chromosomal rearrangements, we acknowledge that we were unable to prioritize to perform an analysis of chromosomal integrity for these embryos, and this should be considered a limitation. Additionally, we recognize that certain malformations, particularly those affecting internal organs, may not be visible to us during dissection.

However, it is important to note that the sections reported in this manuscript were derived from the same lungs previously analyzed by mRNA sequencing and spatial transcriptomics. The protein expression results we observed were consistent with RNA analysis from over 10 embryos, including a pair of twins (Sountoulidis *et al.* 2023)³. Given this consistency across multiple embryos, it is highly unlikely that our protein-based results reflect abnormal lung developmental events due to chromosomal defects.

To address this point, we revised the related sentence in Methods section as follows to more accurately reflect our observation:

Page 18, line 20:

“No major malformations were observed in any of the embryos used in the study.”

³ Sountoulidis, A. et al. A topographic atlas defines developmental origins of cell heterogeneity in the human embryonic lung. Nature Cell Biology 25, 351-365, doi:10.1038/s41556-022-01064-x (2023).

While we recognize the importance of providing a more definitive statement, to address this criticism to the best of our ability and to remain cautious about making claims beyond the scope of our observations and available data, we added a sentence to the Discussion section, where we were already outlining the limitations of this work:

Page 16, line 13:

“Although no malformations were observed in any of these embryos, and our protein-based results were consistent with transcriptomics data across multiple embryos²⁷—making it unlikely that our findings reflect abnormal lung development due to chromosomal defects—another limitation is that these embryos were not karyotyped.”

- It seems that classifying DCs was mainly based on HLA-DR expression, but this is not an exclusive marker of DCs.

HLA-DR is indeed not an exclusive marker for dendritic cells. As explained in our initial rebuttal letter, the annotation of dendritic cells was based on both HLA-DR expression *and* the absence of other lineage markers. To remind, this was also the reason why we refrained from annotating and reporting dendritic cells in the 8.5-week-old sample, because the HLA-DR+ cluster also co-expressed MRC1 in our dataset.

In our revised manuscript, we had clearly discussed this detail/limitation, which we now rephrased again as follows:

Page 15, line 16:

“A more recent study¹⁰ reported the presence of dendritic cells from week 8 onwards using the marker CD1C for immunofluorescence. Although our panel did not include such an exclusive marker for dendritic cells, we were able to annotate them in later weeks based on both HLA-DR expression and the absence of other lineage markers. Thus, we here report the presence of dendritic cells from week 11 onwards, while acknowledging that they may also be present in earlier weeks.”

- Page 5: "Fine" annotation is not appropriate to use as it only gives broad cell types. Classification is only based on 1 or 2 markers.

We agree with the reviewer regarding this detail and rephrased the text accordingly.

Page 6, line 2:

“Next, the data clustered individually for each week was re-annotated for a more detailed analysis of the spatial composition and neighboring patterns of the major cell types at each

developmental week and how these change over the course of early lung development (Supplementary Figure 9-11).”

- The authors state that NK cells and B cells are the least abundant cells during early development. However, the authors’ own single-cell data (Nature Cell Biology, 2023) show that there are plenty of NK cells. So why are they now not seeing NK cells in their data set – perhaps this needs looking at in more detail, using more markers and in more tissue sections?

In response to this comment, we would like to clarify the relevant findings from the Sountoulidis *et al.* (2023)⁴ study mentioned by the reviewer: In that study, 163,000 cells from lung samples spanning 5–14 post-conception weeks (pcw) were sequenced, including 4,113 immune cells. These immune cells included clusters *annotated* as NK cell and B cell clusters. While NK cells were indeed abundant when considering all post-conception weeks cumulatively, the majority of cells in this NK cell cluster were originating from the later developmental stages (11.5–14 pcw), as highlighted in the figure panel we generated from their Supplementary Material. The other transcriptomics study, He *et al.* (2022)⁵ also reported that NK and B cells become more prominent after pcw 15.

from Sountoulidis *et al.* (2023) Nat Cell Biol

⁴ Sountoulidis, A. et al. A topographic atlas defines developmental origins of cell heterogeneity in the human embryonic lung. *Nature Cell Biology* 25, 351-365, doi:10.1038/s41556-022-01064-x (2023).

⁵ He, P. et al. A human fetal lung cell atlas uncovers proximal-distal gradients of differentiation and key regulators of epithelial fates. *Cell* 185, 4841-4860.e4825 (2022). <https://doi.org/10.1016/j.cell.2022.11.005>

In light of these, our general observation and statement of relatively fewer NK and B cells in samples from 6–13 pcw does not contradict the transcriptomic datasets. We still rephrased one related sentence in the Results section as follows:

Page 11, line 9:

“B and NK cells were relatively less abundant immune cell types during early lung development.”

When it comes to NK cells specifically, the limited detection of NK cells in earlier developmental stages (prior to pcw 11) can be attributed to a combination of factors: 1) Random histologic sampling in our analysis (compared to whole-organ dissociation and enrichment of immune cells before sequencing in transcriptomics studies), 2) Potential discrepancies between cell phenotypes that can be confidently resolved, as transcriptomic analysis offers more comprehensive data across the entire transcriptome, while protein markers provide only limited resolution, and 3) Intrinsic differences between RNA and protein expression levels for marker genes used to annotate these cells.

We agree with the reviewer that incorporating additional markers, not only for NK cells but also for other cell types, are necessary for future iterations of this work. This limitation has been acknowledged already in the Discussion section. However, we maintain that the current data serving primarily as a resource for the research community remains in line with the existing transcriptomic datasets, and the findings do not contradict with the expected trends at this developmental stage until more detailed immunophenotyping can be performed.